# MASS: MoErging through Adaptive Subspace Selection

**Donato Crisostomi**[*,1]   **Alessandro Zirilli**[*,1]   **Antonio Andrea Gargiulo**[1]
**Maria Sofia Bucarelli**[1,3,4]   **Simone Scardapane**[1]   **Fabrizio Silvestri**[1]   **Iacopo Masi**[1]
**Emanuele Rodolà**[1,2]

[1]Sapienza University of Rome   [2]Paradigma   [3]CNRS   [4]i3s
{crisostomi, zirilli, gargiulo, masi, rodola}@di.uniroma1.it
simone.scardapane@uniroma1.it   {bucarelli, fsilvestri}@diag.uniroma1.it

## Abstract

Model merging has recently emerged as a lightweight alternative to ensembling, combining multiple fine-tuned models into a single set of parameters with no additional training overhead. Yet, existing merging methods fall short of matching the full accuracy of separately fine-tuned endpoints. We present MASS (*MoErging through Adaptive Subspace Selection*), a new approach that closes this gap by unifying multiple fine-tuned models while retaining near state-of-the-art performance across tasks. Building on the low-rank decomposition of per-task updates, MASS stores only the most salient singular components for each task and merges them into a shared model. At inference time, a non-parametric, data-free router identifies which subspace (or combination thereof) best explains an input's intermediate features and activates the corresponding task-specific block. This procedure is fully training-free and introduces only a two-pass inference overhead plus a $\sim 2\times$ storage factor compared to a single pretrained model, irrespective of the number of tasks. We evaluate MASS on open-vocabulary image classification using ViT-B-16, ViT-B-32 and ViT-L-14 for benchmarks of 8, 14 and 20 tasks respectively, establishing a new state-of-the-art. Most notably, MASS recovers up to $\sim 98\%$ of the average accuracy of individual fine-tuned models, making it a practical alternative to ensembling at a fraction of the storage cost.

○ https://github.com/crisostomi/mass

## 1 Introduction

In the early days of deep learning, the default practice was to train models entirely from scratch. With the rise of massive pretrained networks, research pivoted toward fine-tuning these backbones for specialized tasks (Devlin et al., 2019; Tan et al., 2018; Yosinski et al., 2014; Hu et al., 2022; Radford et al., 2021). Nowadays, with the abundance of fine-tuned models on platforms like HuggingFace[1], we are witnessing a shift toward *no-tuning* methods that leverage both pretrained foundations and diverse fine-tuned endpoints. Among these, *model merging* (Singh & Jaggi, 2020; Ainsworth et al., 2023; Ilharco et al., 2023) has gained significant attention by combining multiple fine-tuned models into a single parameter set, eliminating the need for additional training or data.

An important application of model merging is the combination of models fine-tuned on different tasks that share the same pretrained backbone. Early approaches such as Task Arithmetic (Ilharco et al., 2023) and its extensions (Yadav et al., 2023; Yu et al., 2024; Daheim et al.; Wang et al., 2024; Ortiz-Jiménez et al., 2023; Huang et al., 2024; Akiba et al., 2025) define a task vector as the difference between pretrained and fine-tuned weights, and build a multitask model by summing these vectors to the base model. More recent work (Gargiulo et al., 2025; Daniel et al., 2025) demonstrates that preserving the layer-wise structure of these updates leads to stronger results. Instead of flattening the updates into vectors, methods such as Task Singular Vectors (TSV) (Gargiulo et al., 2025) exploit their matrix form and uncover a clear low-rank structure, where only a small number of singular vectors per task are needed to recover most of the fine-tuned accuracy.

---

[1]https://huggingface.co/docs/hub/models-the-hub

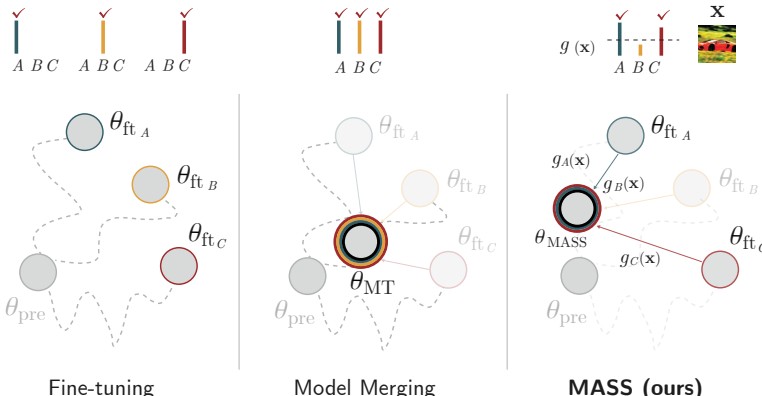

Figure 1: *(left)* Fine-tuning yields three separate models on tasks A, B and C. *(middle)* Model merging produces a single model incorporating the task vectors using a constant function of the input. *(right)* `MASS` stores the pretrained model $\theta_{\text{pre}}$ and the task singular vectors $V^\top$ across tasks. At test time, `MASS` adaptively performs merging using a routing mechanism that chooses appropriate task vectors for the input $\mathbf{x}$, using a thresholded gating function $g(\mathbf{x})$. The gate is the residual between the activations of $\mathbf{x}$ and their projections onto the span of the right singular vectors $V_\perp$.

Despite the progress in performance, most merging methods remain limited by an unrealistic assumption: that the task identity is known at inference time. We challenge this assumption as impractical and argue that under such a setting, the optimal strategy would be compression. Indeed, the low-rank compression method `TSV-C` (Gargiulo et al., 2025) achieves 99.5% normalized accuracy across all benchmarks while requiring only $2\times$ storage and no additional inference overhead, effectively solving the problem. Mixture-of-Experts Merging (MoErging) (Yadav et al., 2024) methods such as `SMILE` (Tang et al., 2024a), `WeMoE` (Tang et al., 2024b), and `TwinMerging` (Lu et al., 2024) go a step further by incorporating a router into the merging process.

However, *these methods still assume that the correct classification head is provided at test time*, thereby inheriting the same unrealistic constraint and failing to exploit the full potential of routing-based approaches.

**Key assumptions.** In this work we assume that the task is *not* known at inference time, and that both the most suitable encoder subspaces and the corresponding classification head must be determined *automatically*. This setting is simultaneously more challenging and more realistic, enabling a single generalist model to handle all tasks encountered during fine-tuning without external supervision.

**Contributions.** We propose `MASS`, a novel MoErging method that dynamically selects the most relevant tasks and corresponding label spaces via an adaptive routing mechanism that conditions the merging process on the input itself (Fig. 1). `MASS` sidesteps the data and training required to train a router by leveraging a novel weight-space router that identifies the most relevant task subspaces as defined by their task singular vectors. By selectively integrating these subspaces into the pretrained backbone and extending the routing decisions to the classification heads, `MASS` enables a training-free and data-free merging process without relying on oracle knowledge of the task at hand.

We evaluate `MASS` on `ViT-B-32`, `ViT-B-16`, and `ViT-L-14` across up to 20 vision tasks and 8 language tasks, demonstrating substantial gains over existing methods. For a modest increase in computational cost ($\sim 2\times$ forward passes) and storage ($\sim 2\times$ parameter footprint regardless of the number of tasks), our method achieves up to 98% of the average accuracy of individual fine-tuned models while handling the full union of expert label spaces without oracle guidance at test time.

Wrapping up, our contribution is three-fold:

- We introduce `MASS`, a MoErging method that augments singular-vector-based merging with adaptive routing, eliminating the need for test-time task knowledge.

- We design a projection-based router that operates without task data or additional tuning, making it directly applicable to the merging setting.

- We present extensive experiments that establish new state-of-the-art results across benchmarks and further provide an interpretation of task singular vectors in text space.

We release our code, checkpoints, and all relevant logs for research purposes.

## 2 BACKGROUND

**Task Vectors** `Task Arithmetic` (TA) (Ilharco et al., 2023) defines task vectors capturing the specific differences in model weights for individual tasks. Formally, the weights $\theta_{\mathrm{MT}}$ of a multi-task model for $T$ tasks are computed by aggregating the task-specific weight differences, or task vectors, as $\theta_{\mathrm{MT}} = \theta_{\mathrm{pre}} + \alpha \sum_{i=1}^{T} \tau_i$, where $\theta_{\mathrm{pre}}$ is the set of pretrained model weights, $\alpha$ is a scaling factor, and $\tau_i = \theta_{\mathrm{ft}_i} - \theta_{\mathrm{pre}}$ is the task vector for task $i$, with $\theta_{\mathrm{ft}_i}$ being the fine-tuned weights for the task. Following Gargiulo et al. (2025), however, we consider these operations at the layer level. From a layer-wise perspective, task arithmetic can be rewritten as $\theta_{\mathrm{MT}}^{(\ell)} = \theta_{\mathrm{pre}}^{(\ell)} + \alpha \sum_{i=1}^{T} \Delta_i^{(\ell)}$, where $\theta_{\mathrm{pre}}^{(\ell)}$ encodes the pretrained weights for layer $\ell$, and $\Delta_i^{(\ell)} = \theta_{\mathrm{ft}_i}^{(\ell)} - \theta_{\mathrm{pre}}^{(\ell)}$ is the task-specific weight difference for task $i$ at layer $\ell$. When layer $\ell$ has a matrix structure, its corresponding $\Delta_i^{(\ell)}$ is called the *per-layer task matrix* for task $i$. The layer index will be omitted for brevity.

**Task Singular Vectors** Given a task $i$, Gargiulo et al. (2025) consider the SVD of the corresponding task matrices $\Delta_i$ on a generic layer $\Delta_i = U_i \Sigma_i V_i^{\top}$. They then perform a low-rank approximation of the $\Delta$s and orthogonalize their singular vectors to reduce inter-task interference. In practice, this is equivalent to summing the top-$k$ rank-one matrices for each task, with an added orthogonalization step to prevent singular vectors belonging to different tasks from interfering. The full procedure is detailed in Algorithm 2 (lines 10–20).

## 3 APPROACH

Our approach is best understood as a pre-processing step followed by an inference-time step. The former consists of a one-time merging procedure to obtain an encoder model $\theta_{\mathrm{MT}}$, as detailed in Algorithm 2. We refer to this as the 'fixed' merging step, as it is performed only once and remains independent of the input. During inference, $\theta_{\mathrm{MT}}$ is used in a dynamic process, outlined in Algorithm 1, consisting of 4 steps:

(i) **First pass**: forward the input through $\theta_{\mathrm{MT}}$ and extract its embedding $\mathbf{z}_\ell$ at a chosen layer $\ell$;

(ii) **Routing**: project $\mathbf{z}_\ell$ onto the task subspaces, selecting those having lowest projection error;

(iii) **Adaptive merge**: merge the selected task subspaces into $\Delta_{\mathrm{ada}}$;

(iv) **Second pass**: classify the input using the final merged model $\theta_{\mathrm{MT}} = \theta_{\mathrm{pre}} + \alpha \Delta_{\mathrm{ada}}$.

### 3.1 FIXED MERGING

We begin with a one-time merging step to produce a model capable of task discrimination. This model provides the intermediate activations used by the router in the first pass. For this, we use `TSV-M` (Gargiulo et al., 2025) due to its subspace-aware aggregation. Although one could alternatively rely on the pretrained base, Table 3 shows that it leads to lower task classification accuracy.

### 3.2 INTEGRATING ROUTING

We extend the aggregation step in Section 2 to include a routing mechanism. Given an input $\mathbf{x}$, the merged model can be adaptively determined by:

$$\theta_{\mathrm{MT}} = \theta_{\mathrm{pre}} + \alpha \sum_{i=1}^{T} \mathbb{1}_{[g_i(\mathbf{x})=1]}(\mathbf{x})\tau_i = \theta_{\mathrm{pre}} + \alpha \sum_{i=1}^{T} \mathbb{1}_{[g_i(\mathbf{x})=1]}(\mathbf{x}) \sum_{j=1}^{k} \sigma_j^i u_j^i v_j^{i\top}, \quad (1)$$

where $g_i(\mathbf{x})$ is a per-task gating function that adaptively selects which task subspaces to activate, and subsequently merge, depending on the input at hand.

---

**Algorithm 1** Adaptive Merging Step

---

**Require:** Pretrained model weights $\theta_{\text{pre}}$, task-specific updates $\{\Delta_i\}_{i=1}^T$, fixed merged model $\theta_{\text{MT}}$, top-$k$ parameter $k$, threshold $\eta$, task-specific classification heads $\{h_i\}_{i=1}^T$, sample $\mathbf{x}$
**Ensure:** Predicted class $c^*$
 1: $\mathbf{z}_\ell \leftarrow \text{ForwardPass}(\theta_{\text{MT}}, \mathbf{x})$        # first pass
 2: **for** $i = 1, \ldots, T$ **do**
 3:     $r_i \leftarrow \|\mathbf{z}_\ell - V_i V_i^\top \mathbf{z}_\ell\|_2$        # residual as per Section 3.2.1
 4: **end for**
 5: $w \leftarrow \text{softmax}(-r)$
 6: $\Omega \leftarrow \{i : w_i \geq \eta\}$        # Select tasks above threshold
 7: $\Omega \leftarrow \text{TopK}(\Omega, w, k)$        # Keep only top-$k$ weighted tasks
 8: **Merge selected subspaces:**    $\Delta_{\text{ada}} \leftarrow \sum_{i \in \Omega} U_i \Sigma_i V_i^\top$
 9: **Compute adaptive model:**    $\theta_{\text{MASS}} \leftarrow \theta_{\text{pre}} + \alpha \Delta_{\text{ada}}$
10: **Classification procedure**
11: $\mathbf{z}_{L-1} \leftarrow \text{ForwardPass}(\theta_{\text{MASS}}, \mathbf{x})$        # Compute shared representation
12: $\mathbf{z}_i \leftarrow h_i(\mathbf{z}_{L-1})$        # Evaluate each head
13: $(i^\star, c^\star) \leftarrow \underset{(i,c) \in \Omega \times \{1, \ldots, C_i\}}{\arg\max} z_i[c]$        # Highest logit across heads
14: **return** $c^\star$

---

Traditionally, however, routers require either task-specific *data* for non-parametric procedures such as nearest-neighbor-based routing, or both data and *training* for parametric routers. This contrasts with the typical merging scenario, which does not assume access to the data used to train the endpoint models, as these could, in principle, be downloaded from any public model repository. We therefore introduce a completely ***data-* and *tuning*-free** approach.

### 3.2.1 PROJECTION-BASED ROUTING

Given an input intermediate representation $\mathbf{z}_\ell$ for a predetermined layer $\ell$, we want to identify which task subspace (or set of subspaces) is the most relevant. Concretely, one way to do this is to compute the Euclidean residual of $\mathbf{z}_\ell$ after projecting onto $\text{span}(V_i^{(\ell)})$:

$$r_i = \left\| \mathbf{z}_\ell - \text{Proj}_{V_i^{(\ell)}}(\mathbf{z}_\ell) \right\|_2 \quad ,$$

where $\text{Proj}_{V_i^{(\ell)}}(\mathbf{z}_\ell) = V_i^{(\ell)} (V_i^{(\ell)})^\top \mathbf{z}_\ell$ is the orthogonal

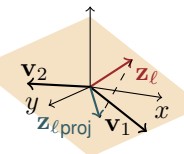

Figure 2: Projection of the activations $\mathbf{z}_\ell$ onto the span of TSVs $\mathbf{v}_1, \mathbf{v}_2$.

projection of $\mathbf{z}_\ell$ onto $\text{span}(V_i^{(\ell)})$, which yields the minimum-error ($L_2$-optimal) reconstruction within that subspace (see Proposition D.1). At this point, the additive inverse of the residual vector $\mathbf{r} = r_0, \ldots, r_T$ is normalized through a softmax to obtain the coefficients. The router then picks those exceeding a predetermined threshold $\eta$, limiting the selection to the top-$k$ when more tasks surpass it. For details regarding the choice of the layer used to compute the residual, see Section 4.2.

### 3.2.2 ACCOUNTING FOR REDUNDANT DIRECTIONS

For projection-based routing to be effective, no task should overshadow the others. Consider, for example, three tasks: MNIST ⬛, EMNIST ⬛, and KMNIST ⬛. Being trained on very similar datasets covering the same classes, $\Delta_{\text{MN}}$ and $\Delta_{\text{EMN}}$ share a large portion of their right-singular directions: $\text{span}(V_{\text{MN}}^{(\ell)}) \approx \text{span}(V_{\text{EMN}}^{(\ell)})$, while KMNIST has some distinct directions $V_{\text{KM}}^{(\ell)}$ capturing more Japanese *kana*-like shapes. However, all three tasks may agree on certain generic "black background, white glyph" features. Because MNIST and EMNIST partially *reinforce* these directions (they both include them), the *union* of their subspaces can appear "wider" or more dominant in that region of feature space. Consequently, for many test samples $\mathbf{z}_\ell$ with black backgrounds and centered shapes:

$$\left\| \mathbf{z}_\ell - \text{Proj}_{V_{\text{MN}} \cup V_{\text{EMN}}}(\mathbf{z}_\ell) \right\|_2 < \left\| \mathbf{z}_\ell - \text{Proj}_{V_{\text{KM}}}(\mathbf{z}_\ell) \right\|_2.$$

Hence, the router sees a smaller residual for MNIST/EMNIST, declaring those tasks more suitable even if the glyph belongs to KMNIST.

During the fixed merging step, instead of aggregating all task matrices, we only keep those that are *sufficiently distinct*. We first select a single task matrix as the initial element of the merge set. Then, for each remaining task matrix, we determine whether to include it based on its similarity to the matrices already in the set. A task matrix is added only if its similarity to all previously accepted matrices remains below a predefined threshold.

Formally, let $\{\Delta_{a_1}, \ldots, \Delta_{a_r}\}$ be the set of accepted updates at a given layer. When evaluating a new task update $\Delta_i$, we flatten it as $\delta_i = \mathrm{vec}(\Delta_i)$ and compute its cosine similarity $\mathrm{sim}(\delta_i, \delta_{a_m})$ with each previously accepted update $\delta_{a_m}$. If

$$\max_{1 \le m \le r} \mathrm{sim}(\delta_i, \delta_{a_m}) > \varepsilon,$$

where $\varepsilon$ is a user-specified threshold (e.g. $\varepsilon = 0.3$), we discard $\Delta_i$ and do *not* merge it; else we accept it. This ensures that highly similar task subspaces do not overshadow less common ones. This procedure is performed before merging, see lines 2–8 of Algorithm 2.

### 3.3 ADAPTIVE MERGING AND INFERENCE

With the router having selected a subset $\Omega$ of relevant tasks, we merge their subspaces via `TSV-M` (Gargiulo et al., 2025) into a single model $\theta_{\mathrm{MASS}}$. Crucially, *unlike standard model merging, which assumes an oracle provides the correct task head at inference, we do not know the task in advance.* Instead, after obtaining the shared representation $\mathbf{z}_{L-1} \in \mathbb{R}^d$ from $\theta_{\mathrm{MASS}}$, we run each head $h_i : \mathbb{R}^d \to \mathbb{R}^{C_i}$ for every selected task $i \in \Omega$:

$$\mathbf{z}_i = h_i(\mathbf{z}_{L-1}), \quad \mathbf{z}_i \in \mathbb{R}^{C_i}.$$

We then pick the highest logit among all heads in $\Omega$. Formally:

$$(i^\star, c^\star) = \underset{(i,c) \in \Omega \times \{1,\ldots,C_i\}}{\arg\max} z_i[c].$$

In other words, we identify which head $i^\star$ is most "confident" and select its predicted class $c^\star$. *This procedure covers the unknown-task scenario*, allowing the model to determine both head and label space on a per-input basis.

### 3.4 RESIDUAL MINIMIZATION AS MAXIMUM A POSTERIORI ESTIMATION

The task selection process in `MASS` can be viewed as a *maximum a posteriori* (MAP) estimation problem. If residuals follow an isotropic Gaussian, the likelihood of a feature vector given a task decays exponentially with its squared $\ell_2$ reconstruction error. Thus, choosing the task with minimal residual is equivalent to the MAP estimate. This view parallels probabilistic PCA (Tipping & Bishop, 1999), where squared reconstruction error minimization corresponds to maximum likelihood estimation under the same Gaussian assumption. Residual-based selection is therefore statistically optimal under a simple, least-informative model that is particularly apt for `MASS`, which lacks training data to fit more complex distributions. The isotropic prior treats all directions equally, avoiding bias toward specific tasks.

**Proposition 3.1.** *Let $\mathbf{z}_\ell \in \mathbb{R}^d$ be a feature vector, and for each task $i$, decompose it as*

$$\mathbf{z}_\ell = V_i V_i^\top \mathbf{z}_\ell + \varepsilon_i, \qquad \varepsilon_i = \left(I - V_i V_i^\top\right)\mathbf{z}_\ell.$$

*Assume $\varepsilon_i \sim \mathcal{N}(0, \sigma^2 I)$ and a uniform prior over tasks: $p(task = i) = \frac{1}{K}$ for all $i \in \{1, 2, \ldots, K\}$.. Then the maximum a posteriori estimate of the task reduces to*

$$\hat{i}_{\mathrm{MAP}} = \arg\max_i p(task = i \mid \mathbf{z}_\ell) = \arg\min_i \|\varepsilon_i\|_2^2.$$

## 4 EXPERIMENTS

### 4.1 MERGING PERFORMANCE

**Models, baselines and datasets** We conduct our experiments on three versions of the `CLIP` (Radford et al., 2021) model, each equipped with a different `ViT` (Dosovitskiy et al., 2021) visual

| | Method | ViT-B-32 | | | ViT-B-16 | | | ViT-L-14 | | |
|---|---|---|---|---|---|---|---|---|---|---|
| | | 8 tasks | 14 tasks | 20 tasks | 8 tasks | 14 tasks | 20 tasks | 8 tasks | 14 tasks | 20 tasks |
| Base | Zeroshot | 48.1(54.8) | 56.9(64.5) | 57.5(65.2) | 55.3(60.6) | 61.9(67.9) | 62.5(68.3) | 64.9(69.2) | 69.1(73.8) | 68.2(72.7) |
| Base | Finetuned | 90.3(100) | 89.0(100) | 89.5(100) | 92.4(100) | 91.3(100) | 91.9(100) | 94.2(100) | 93.4(100) | 94.0(100) |
| Fixed | Weight Averaging | 67.1(74.6) | 65.5(73.8) | 64.4(72.6) | 73.0(78.8) | 70.8(77.3) | 69.2(75.3) | 80.5(85.2) | 78.5(83.8) | 76.1(80.9) |
| Fixed | Task Arithmetic | 68.8(75.7) | 64.6(72.5) | 64.0(71.9) | 73.0(78.3) | 70.6(77.0) | 69.0(75.0) | 84.4(89.3) | 80.4(85.8) | 76.9(81.7) |
| Fixed | Consensus TA | 72.6(80.1) | 70.3(78.9) | 68.5(76.8) | 75.9(81.7) | 74.9(81.7) | 72.2(78.4) | 85.5(90.5) | 82.0(87.6) | 78.9(83.8) |
| Fixed | TSV-M | 83.2(91.8) | 78.6(88.0) | 75.6(84.3) | 85.5(92.2) | 81.4(88.8) | 78.8(85.5) | 91.2(96.7) | 88.8(94.9) | 87.5(93.0) |
| Fixed | Iso-C | 82.8(91.7) | 78.4(88.0) | 73.2(81.9) | 87.5(94.4) | 79.8(87.0) | 75.3(81.6) | 92.6(98.2) | 89.6(95.8) | 86.8(92.3) |
| Fixed | Iso-CTS | 82.0(90.9) | 80.6(90.4) | 77.0(86.2) | 88.7(95.9) | 84.1(91.8) | 80.7(87.7) | 92.8(98.5) | **91.1(97.4)** | 89.2(94.9) |
| MoE | WeMoE | **88.8(97.5)** | 74.3(82.8) | 68.2(76.3) | 89.1(96.4) | 76.6(83.2) | 65.0(70.5) | 88.7(94.2) | 72.3(76.8) | 65.0(69.4) |
| MoE | SMILE-1 | 83.2(92.1) | 75.4(84.5) | 72.8(82.3) | 87.8(94.9) | 81.7(89.5) | 79.5(86.7) | 91.2(96.7) | 86.6(92.7) | 84.9(90.5) |
| MoE | SMILE-2 | 84.4(93.5) | 76.4(85.6) | 74.1(83.8) | 89.0(96.2) | 82.7(90.7) | 80.4(87.7) | 92.0(97.6) | 87.1(93.4) | 85.5(91.1) |
| MoE | **MASS** | 87.0(96.5) | **82.9(93.2)** | **81.1(90.9)** | **90.6(98.0)** | **87.8(96.1)** | **81.1(88.7)** | **92.9(98.6)** | 90.9(97.3) | **90.8(96.6)** |

Table 1: Average absolute accuracy results on model merging benchmarks; subscript (in parentheses) is the normalized average accuracy.

| Method | CoLA | MNLI | MRPC | QNLI | QQP | RTE | SST-2 | STS-B | Avg. |
|---|---|---|---|---|---|---|---|---|---|
| Finetuned | 75.0(100) | 83.4(100) | 87.5(100) | 91.5(100) | 85.4(100) | 85.9(100) | 93.6(100) | 88.7(100) | 86.4(100) |
| Weight Averaging | 69.1(92.1) | 62.6(75.1) | 79.4(90.7) | 89.8(98.1) | 83.9(98.2) | 81.2(94.5) | 91.7(97.9) | 73.2(82.5) | 78.9(91.3) |
| Task Arithmetic | 70.5(94.0) | 57.8(69.3) | 78.4(89.6) | 90.2(98.6) | 83.6(97.9) | 80.5(93.7) | 92.3(98.6) | 77.8(87.7) | 78.9(91.3) |
| Ties-Merging | 70.3(93.7) | 65.0(77.9) | 78.9(90.2) | 90.2(98.6) | 83.5(97.8) | 81.6(95.0) | 91.7(97.9) | 78.3(88.3) | 79.9(92.5) |
| WeMoE | 72.5(96.6) | 79.0(94.7) | 51.9(59.3) | 89.3(97.5) | 69.6(81.4) | 81.5(94.8) | 88.6(94.6) | 82.1(92.5) | 76.8(88.9) |
| SMILE-1 | 72.0(96.0) | 84.2(101.0) | 84.3(96.3) | 91.3(99.8) | 84.7(99.2) | 84.1(97.9) | 93.3(99.7) | 87.0(98.1) | 85.1(98.5) |
| SMILE-2 | 73.2(97.6) | **84.2(101.0)** | 85.0(97.1) | **91.3(99.8)** | 84.9(99.4) | 84.8(98.7) | 93.5(99.9) | 87.3(98.4) | 85.5(99.0) |
| **MASS** | **74.1(98.8)** | 83.2(99.8) | **85.8(98.1)** | 90.9(99.3) | **85.1(99.6)** | 84.9(98.8) | **94.2(100.6)** | **88.9(100.2)** | **85.9(99.4)** |

Table 2: Accuracy when merging 8 fine-tuned models on the GLUE (Wang et al.) benchmark; Normalized average accuracy in subscript.

encoder: `ViT-B-32`, `ViT-B-16`, and `ViT-L-14`. As baselines, we compare against multiple training-free merging strategies, notably weight averaging, `Task Arithmetic` (Ilharco et al., 2023), and `Consensus Merging` (Wang et al., 2024). For additional context, zero-shot performance serves as a null reference point, and the mean accuracy of individually fine-tuned models marks the upper bound on achievable performance. We evaluate on three collections of tasks, containing 8, 14, and 20 tasks, respectively. The latter is the most extensive setup considered in (Wang et al., 2024; Gargiulo et al., 2025; Daniel et al., 2025). We refer to Section B.3 for the specifics. We quantify results using both average absolute accuracy and average normalized accuracy.

**MoErging results** Table 1 reports the average absolute accuracy and the corresponding normalized accuracy (subscript, also in percentage) for each method, model size, and number of vision tasks. To adapt SMILE and WeMoE to our general setting, we employ a majority-voting-based heuristic to route the classification head. Details in §B.6.

We first note that MASS sets a new state of the art for MoErging across 8 out of 9 benchmarks, with gains as high as $\approx 6\%$ with respect to the best performing baseline. Figure 3 shows a per-dataset breakdown of the results, indicating that the accuracy is consistently high throughout all the datasets and not skewed on a subset of easier ones. In terms of storage, MASS requires a constant $2\times$ parameter increase, whereas other MoErging baselines range from $\sim 2.5\times$ up to $\sim 14\times$. A breakdown is provided in §B.1.

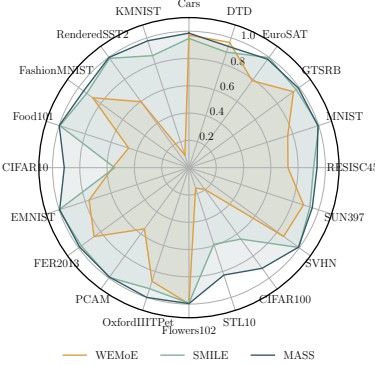

Figure 3: Per-dataset accuracies.

We also report the accuracies obtained by fixed merging methods, *i.e.* those not employing a router. These are evaluated with the ground truth classification head, *i.e.* assuming the task is known at

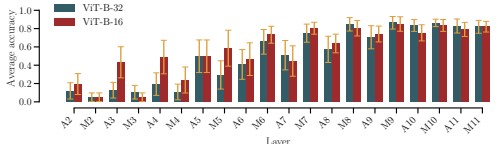
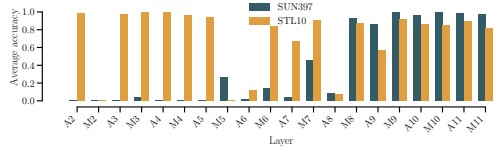

(a) Averaged across all tasks for two models.

(b) Focusing on `SUN397` and `STL10` for a `ViT-B-32`.

Figure 4: Per-layer task accuracies for `ViT-B-32` on the 20-task benchmark. Layers starting with 'A' indicate attention layers, while those starting with 'M' refer to MLPs.

inference time. While our evaluation setting is significantly more challenging, `MASS` still outperforms these ones by a consistent margin. In particular, when comparing it with the `TSV-M` baseline it builds upon, we see that routing produces a $\approx 5\%$ increase in accuracy.

We further compare `MASS` with `TwinMerging` (Lu et al., 2024) on the 8-task benchmark using `ViT-B-32`, the only vision setting with reported results in their paper. Despite the lack of evaluation code for other setups, we find that `MASS` achieves **97.6%** normalized accuracy in this shared setting, exceeding the 95.3% reported for `TwinMerging`. Notably, the latter assumes oracle knowledge of the correct classification head, whereas `MASS` selects both the merged experts and the output space **at inference time**. Despite operating in a more general setting, `MASS` still reaches a higher accuracy. We also report in Table 2 the results on merging 8 `Flan-T5-Base` models (Chung et al., 2024) finetuned on the language tasks in GLUE (Wang et al.). `MASS` still yields the best results, showing its benefits to be modality agnostic. MASS obtains the highest absolute accuracy on 5 out of the 9 GLUE tasks (CoLA, MRPC, QQP, RTE, SST-2, STS-B), while performing slightly below the best baseline on the remaining ones (MNLI, QNLI). We observe that the two tasks where MASS does not achieve the top score are the NLI-style benchmarks (MNLI and QNLI). This is consistent with these tasks requiring more broad semantic reasoning rather than localized feature shifts, possibly resulting in more diffuse and higher-rank task vectors compared to other GLUE tasks.

**Batched inference** In the main experiments, we evaluated our method in the most challenging scenario, where each sample may belong to a different task, forcing the router to operate per input. Yet, in many practical settings (e.g., batched requests from the same domain), several inputs share the same task. In such cases, we can router-select a single merged model once per batch, closing the gap almost entirely: as shown in Figure 5, our approach achieves a mean normalized accuracy of at least 97% in 8 out of 9 settings, effectively matching individually fine-tuned models. This indicates that while our per-sample approach is already effective, batching can further reduce overhead and significantly boost accuracy.

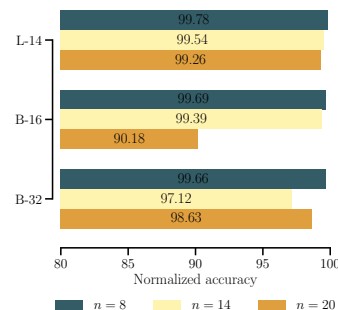

Figure 5: Batched accuracy.

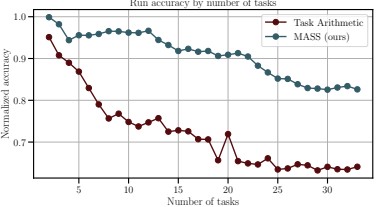

Figure 6: Merging accuracy when scaling the number of tasks.

**Scaling the number of tasks** We now study how `MASS` scales as the number of tasks increases. For this, we incorporate 13 additional datasets and evaluate all merges from 2 up to 33 tasks. Fig. 6 shows that `MASS` maintains accuracy even on the largest task set. Interestingly, the accuracy is not a monotonic function of the number of tasks. In several regions the performance even increases when adding more tasks, suggesting that scalability is driven more by the composition of the task set than by its cardinality alone. Overall, `MASS` maintains high accuracy throughout the entire range of tasks, in clear contrast to the steep degradation observed with TA.

## 4.2 CHOOSING A ROUTING LAYER

We now investigate the effect of routing layer selection on task accuracy. Figure 4a shows the task prediction accuracy obtained by routing at different layers for `ViT-B-32` and `ViT-B-16` architectures. Interestingly, the best layer is consistent across architectures, with `ViT-B-32` and `ViT-B-16` both achieving peak performance at layer 9, and MLP layers exhibiting slightly better performance than the self-attention layers. However, it is immediately clear that the best layer varies significantly across tasks. In fact, some layers exhibit a variance of up to $40\%$ in accuracy across tasks. This can be better appreciated by looking at Figure 4b, where we can see the per-layer task accuracies on `STL10` and `SUN397` for a `ViT-B-32`. While both tasks can be accurately predicted with the best chosen layer $\ell = 9$, the former shows a marked improvement in accuracy when routing at earlier layers $\ell = 3, 4, 5$, while the latter benefits from routing at later layers $\ell = 9, 10, 11$ and results in poor performance in the earlier ones. This suggests that the best routing layer is task-dependent, and may encourage further research on adaptive ways to determine it. We report an analogous analysis for the `ViT-L-14` model in the appendix.

## 4.3 COMPARISON WITH OTHER ROUTERS

We compare our router with two alternatives that differently balance accuracy, data, and compute.

Nearest Neighbor (`nn`) first constructs a small *support set* of representative examples from each task's validation data. For each test sample, we compare its intermediate representation $\mathbf{z}_\ell$ to the stored representations of all support examples. Formally, we compute the cosine similarity to each support sample and select the nearest neighbor among all tasks, inferring the task identity from the task to which the support sample belongs. This method requires no additional parameters but assumes access to (and storage for) a curated batch of validation embeddings per task.

The second approach (`mlp`) fits an MLP over the union of validation sets from all tasks, with each example labeled by its originating task. After extracting $\mathbf{z}_\ell$, we train an MLP $f_\theta$ to predict the task via cross-entropy loss. Full architectural details are in the supplementary materials.

**Results** Table 3 shows the average normalized accuracy obtained by MASS when varying routing strategy. The `nn` approach, which stores and compares with a small validation set from each task, generally performs well but remains slightly below our best results. The MLP router achieves the highest accuracy overall, but it relies on a labeled validation set for training–a requirement that contradicts the core premise of merging methods, where access to original task data is often unavailable. This dependency makes the MLP less broadly applicable in realistic scenarios, such as when downloading fine-tuned models from public hubs without any accompanying data.

| MASS | ViT-L-14 | | |
|---|---|---|---|
| + | 8 tasks | 14 tasks | 20 tasks |
| nn | 94.0 | 92.1 | 92.0 |
| mlp | 98.9 | 99.5 | 98.3 |
| proj$_{\text{PRE}}$ | 99.1 | 97.7 | 91.9 |
| proj$_{\text{TSV-M}}$ | 98.6 | 97.3 | 96.6 |

Table 3: Average normalized accuracy for different routers.

Focusing on the projection-based router (`proj`), we observe that starting from the `TSV-M` model (proj$_{\text{TSV-M}}$) outperforms routing from the pretrained backbone (proj$_{\text{PRE}}$) when scaling the number of tasks to 20. This is further confirmed in Table 15, where the gap widens to $\approx 10\%$ accuracy for a `ViT-B-32`. This gap underscores the core key insight behind our method: `TSV-M` arranges each task's top singular vectors into distinct, orthogonal subspaces, all contained in the final merged model. Therefore, proj$_{\text{TSV-M}}$ only needs to measure how well each subspace reconstructs the activations, "finding back" the subspace that was originally embedded for each task. In other words, once `TSV-M` has embedded each task's directions into a single model, a projection is enough to pinpoint the correct subspace, requiring no labels or additional training.

## 4.4 INTERPRETING TASK SINGULAR VECTORS

Finally, we attempt to interpret the task singular vectors that are used for routing in MASS. Notably, prior work suggests that mid-to-late layer embeddings in CLIP-like models preserve semantic content, implying that routing at this layer hinges on meaningful features (Gandelsman et al., 2024). To validate this, we interpret TSVs using the TEXTSPAN algorithm (Gandelsman et al., 2024; Basile et al., 2024). The latter iteratively identifies and removes the most influential text directions, revealing which concepts best explain how the model's weight changes affect its representation. Impor-

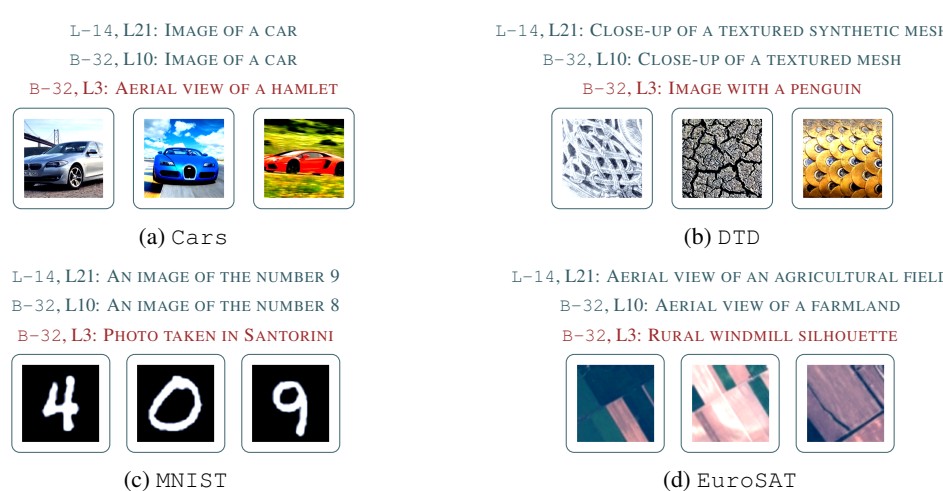

L-14, L21: IMAGE OF A CAR
B-32, L10: IMAGE OF A CAR
B-32, L3: AERIAL VIEW OF A HAMLET

(a) Cars

L-14, L21: CLOSE-UP OF A TEXTURED SYNTHETIC MESH
B-32, L10: CLOSE-UP OF A TEXTURED MESH
B-32, L3: IMAGE WITH A PENGUIN

(b) DTD

L-14, L21: AN IMAGE OF THE NUMBER 9
B-32, L10: AN IMAGE OF THE NUMBER 8
B-32, L3: PHOTO TAKEN IN SANTORINI

(c) MNIST

L-14, L21: AERIAL VIEW OF AN AGRICULTURAL FIELD
B-32, L10: AERIAL VIEW OF A FARMLAND
B-32, L3: RURAL WINDMILL SILHOUETTE

(d) EuroSAT

Figure 7: Captions obtained by decoding task singular vectors as text as described in Section 4.4, accompanied by task representative images. Captions produced by the task singular vectors of predictive layers reflect the task content, those obtained by non-predictive ones do not.

tantly, while TEXTSPAN was originally developed to analyze image embeddings, we employ it here to interpret singular vectors derived from weight updates.

Our results, illustrated in Figure 7, confirm that TSVs capture meaningful and interpretable visual-language associations. For example, the singular vectors associated with the Cars dataset (Krause et al., 2013) strongly activate the description "IMAGE OF A CAR", whereas those for the DTD dataset (Cimpoi et al., 2014) align closely with the phrase "CLOSE-UP OF A TEXTURED MESH". Interestingly, across architectures (ViT-L-14, ViT-B-32), we find consistent textual concepts emerging from similar singular vectors for the same tasks. For instance, the singular vectors for both ViT-L-14 at layer 21 and ViT-B-32 at layer 10 align with almost equal phrases. This consistency hints at the intriguing possibility of transferring semantic information across architectures using text embeddings as an interpretable common ground.

We further note that the interpretability of these embeddings strongly depends on the chosen routing layer. While mid-to-late layers (e.g., layer 10 in ViT-B-32 and layer 21 in ViT-L-14) yield semantically meaningful descriptions aligned closely with each dataset's visual domain, earlier layers (e.g., layer 3 in ViT-B-32) produce irrelevant or misleading concepts such as "IMAGE WITH A PENGUIN" for the texture dataset (DTD) or "PHOTO TAKEN IN SANTORINI" for digit classification (MNIST). Such discrepancies suggest that early layers encode generic or low-level features, while mid-to-late layers become progressively specialized toward domain-specific semantic structures.

Overall, these analyses validate our routing strategy: the task singular vectors at mid-layer embeddings capture precisely the semantic differences the router exploits to discriminate tasks. This also points to a deeper insight: the singular vectors derived from weight updates mirror the semantic structure of the data itself, highlighting a direct connection between weights and data.

## 5 RELATED WORK

**Model Merging** has emerged as a lightweight alternative to ensembling. Early work focused on aligning models trained from scratch with different random seeds by solving permutation and mode-connectivity issues (Frankle et al., 2020; Entezari et al., 2022; Ainsworth et al., 2023; Crisostomi et al., 2025). More recent approaches instead merge multiple fine-tuned models derived from a common backbone (Ilharco et al., 2023; Yadav et al., 2023; Yu et al., 2024; Wang et al., 2024; Gargiulo et al., 2025; Daniel et al., 2025). These methods typically treat task updates as vectors and combine them through rescaling, pruning, or averaging, while newer work shows that respecting the layer-wise structure of updates significantly improves results (Gargiulo et al., 2025; Daniel et al., 2025). Our method builds directly on this line by adding adaptivity through input-driven routing, narrowing the performance gap between merged and fine-tuned models.

**MoErging.** A parallel line of work incorporates routing into merging, often referred to as "Mo-Erging" (He et al., 2023; Yadav et al., 2024; Tang et al., 2024a; Lu et al., 2024). In the LLM domain, routers are trained to select among LoRAs or adapters (Jang et al., 2023; Chronopoulou et al., 2023; Muqeeth et al., 2024), sometimes requiring additional supervision or fine-tuning. Closer to our setting, `Transformer`$^2$ (Sun et al., 2025) introduces a two-step routing pipeline but relies on a modified fine-tuning procedure, while `SMILE` (Tang et al., 2024a) proposes a data- and training-free approach that leaves the pretrained backbone unchanged and *adds* task-specific low-rank updates at inference through layer-wise routing. In contrast, `MASS` embeds all updates into a single model, performs task selection once across layers, and *deactivates* irrelevant subspaces via a projection-based router. Finally, `TwinMerging` (Lu et al., 2024) requires per-task labels to train its router, whereas `MASS` is fully training- and data-free. We defer further discussion of related work to Section A.

## 6 CONCLUSIONS

In this paper, we introduced `MASS` (MoErging through Adaptive Subspace Selection), a method that aggregates low-rank task updates with adaptive routing, jointly selecting both encoder subspaces and classification heads without test-time supervision. To address the absence of per-task data in realistic merging scenarios, we designed a projection-based router that is entirely training- and data-free.

Our experiments show that `MASS` achieves state-of-the-art performance, recovering nearly the full accuracy of fine-tuned experts at a fraction of their combined storage cost. Future work includes refining the router for more precise subspace selection and extending `MASS` to out-of-distribution settings, where unseen tasks could be composed on the fly from existing singular vectors.

## 7 ACKNOWLEDGMENTS

This work is supported by the MUR FIS2 grant n. FIS-2023-00942 "NEXUS" (cup B53C25001030001), and partly by Sapienza University of Rome via the Seed of ERC grant "MINT.AI" (cup B83C25001040001). It was also supported by projects PNRR MUR PE0000013-FAIR under the MUR National Recovery and Resilience Plan funded by the European Union - NextGenerationEU, PRIN 2022 project 20227YET9B "AdVVent" CUP code B53D23012830006, and the project BEAT (Better dEep leArning securiTy), a Sapienza University project.

This work has also been supported by the French government, through the 3IA Côte d'Azur Investments in the project managed by the National Research Agency (ANR) with the reference number ANR-23-IACL-0001.

We finally acknowledge ISCRA for awarding this project access to the LEONARDO supercomputer, owned by the EuroHPC Joint Undertaking, hosted by CINECA (Italy).

## REPRODUCIBILITY STATEMENT

All implementation details and hyperparameter settings required to reproduce our results are provided in Sections 3, 4 and in Appendix B. The code is provided in the supplementary materials and will be publicly released along with all checkpoints and logs.

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

# A   EXTENDED RELATED WORK

## A.1   MODEL MERGING

Model Merging has recently gained traction as a computationally efficient alternative to ensembling. Early methods, motivated by linear mode connectivity (Frankle et al., 2020; Entezari et al., 2022; Mirzadeh et al., 2021; Garipov et al., 2018), primarily aligned models trained with different optimization seeds. This was achieved by finding neuron permutations matching these ones before the aggregation (Ainsworth et al., 2023; Jordan et al., 2023; Crisostomi et al., 2025; Singh & Jaggi, 2020; Stoica et al., a; Guerrero-Peña et al., 2023; Navon et al., 2023; Horoi et al., 2024). More recent merging approaches aggregate instead multiple fine-tuned models derived from a shared pre-trained backbone (Ilharco et al., 2023; Yadav et al., 2023; Yu et al., 2024; Matena & Raffel, 2022; Wortsman et al., 2021; Davari & Belilovsky, 2025; Wang et al., 2024; Zhou et al., 2024; Gargiulo et al., 2025; Ortiz-Jiménez et al., 2023; Mencattini et al., 2025; Huang et al., 2024; Daheim et al.; Stoica et al., b; Yang et al., 2024a; Tang et al.). These methods incorporate various strategies to improve the merging, such as finding the optimal combination of task vectors (Yang et al., 2024b), mitigating sign disagreement (Yadav et al., 2023), randomly dropping a fraction of the updates (Yu et al., 2024), or employing evolutionary strategies (Akiba et al., 2025; Mencattini et al., 2025). The most recent line of work considers task vectors at a layer level, accounting this way for the natural structure of the layers and significantly improving the merging outcome (Stoica et al., b; Gargiulo et al., 2025; Daniel et al., 2025). Our approach builds upon the latter methodologies by introducing adaptivity through input-driven routing, significantly narrowing the performance gap between the finetuned endpoints and their resulting merge.

## A.2   MOERGING

Model Merging with Mixture-of-Experts, often referred to as "MoErging" (He et al., 2023; Yadav et al., 2024; Jang et al., 2023; Chronopoulou et al., 2023; Belofsky, 2023; Zhao et al., 2024; Muqeeth et al., 2024; Tang et al., 2024b; Ostapenko et al., 2024; Cheng et al., 2025; Tang et al., 2024a; Lu et al., 2024; Sun et al., 2025), explores how independently trained experts–potentially contributed by a decentralized community–can be combined within a single adaptive model by dynamically selecting which expert(s) should handle a given input. In the LLM domain, specialized modules (e.g., LoRAs or adapters) are merged via parametric or data-driven routers that match the incoming prompt to the most relevant module (Jang et al., 2023; Chronopoulou et al., 2023; Belofsky, 2023; Zhao et al., 2024; Muqeeth et al., 2024; Tang et al., 2024b; Ostapenko et al., 2024; Cheng et al., 2025); some approaches, such as PHATGOOSE (Muqeeth et al., 2024), require fine-tuning additional routing parameters, whereas others, like weight-ensembling MoE (Tang et al., 2024b), may need to train the router at test time. Sharing a similar two-pass pipeline, $\text{Transformer}^2$ (Sun et al., 2025) modifies the finetuning procedure to yield task-aligned singular vectors, allowing for expert routing at test time. Differently from the latter, MASS works on any independently fine-tuned models finetuned with no ad hoc routines. A similar strategy was independently introduced by SMILE (Tang et al., 2024a), which, like our method, enables data-free merging of multiple experts. However, SMILE leaves the pretrained backbone intact and selectively *adds* task-specific low-rank updates at inference, whereas MASS begins with a single model containing all updates and *deactivates* any irrelevant subspaces via a router. The fact that both lines of work arrived at a similar subspace-activation concept, despite differing motivations, highlights the versatility and broad appeal of such an approach. Lastly, TwinMerging (Lu et al., 2024) similarly merges a shared expert and multiple task-specific experts via a gating function. However, TwinMerging relies on flat task arithmetic (Ilharco et al., 2023) and requires per-task labeled data to train its router, reducing its applicability. In contrast, MASS operates in a fully data-free and training-free regime by design.

# B   ADDITIONAL DETAILS

We here describe the details required to implement and reproduce our results. The code is provided in the supplementary materials. In particular, Section B.2 describes implementation details, Section B.4 specifies the employed evaluation metrics, Section B.5 reports the employed architecture and Section C.2 specificies the hyperparameters and how they were chosen.

| | Method | ViT-B-32 | | | ViT-B-16 | | | ViT-L-14 | | |
|---|---|---|---|---|---|---|---|---|---|---|
| | | 8 tasks | 14 tasks | 20 tasks | 8 tasks | 14 tasks | 20 tasks | 8 tasks | 14 tasks | 20 tasks |
| MoE | WeMoE | 5.06 | 8.06 | 11.05 | 5.12 | 8.16 | 11.20 | 5.78 | 9.31 | 12.84 |
| | SMILE-1 | **1.61** | 2.07 | 2.52 | **1.62** | 2.09 | 2.55 | **1.47** | **1.82** | 2.18 |
| | SMILE-2 | 3.05 | 4.60 | 6.14 | 3.09 | 4.67 | 6.24 | 2.59 | 3.78 | 4.96 |
| | **MASS** | 2.00 | **2.00** | **2.00** | 2.00 | **2.00** | **2.00** | 2.00 | 2.00 | **2.00** |

Table 4: Relative parameters increase with respect to the base model.

## B.1 STORAGE AND COMPUTE OVERHEAD

The $\sim 2\times$ **compute** figure is an approximation: we run the backbone twice (fixed TSV-M (Gargiulo et al., 2025) pass + routed pass) and the router itself adds $< 1\%$ FLOPs. Since model-merging saves training and storage rather than inference time, this modest overhead is acceptable. MASS routes per sample, so each incurs this cost; however, batching over the same task can reduce it. For example, with 32 samples, routing costs only $1.06\times$ forward equivalents. The **storage overhead** is exactly $2\times$: alongside the merged model, we store $1/T$ TSVs per task. Summed across tasks and layers, these form a second full set of weights, effectively doubling storage. A full comparison with the overhead incurred by the baselines is provided in Table 4.

## B.2 IMPLEMENTATION

We used the same model checkpoints as Consensus TA (Wang et al., 2024), except for the one for the EMNIST dataset which we had to re-finetune due to an inconsistency in image orientation between EMNIST and MNIST. Shortly, the *torchvision*[2] version yields rotated and flipped images, spuriously yielding extremely similar models (same classes, roughly same dataset statistics) that performed very poorly when interchanged. Simply re-rotating and flipping the EMNIST images to match the orientation of MNIST solves the issue. We further used a single classification head for STL10 and CIFAR10 due to their 9 shared classes. The final head has the shared classes plus the two dataset-specific ones, *i.e.* monkey and frog.

## B.3 BENCHMARKS AND DATASETS

The 8-task benchmark, introduced in (Ilharco et al., 2023), comprises the following datasets: Cars, DTD, EuroSAT, GTSRB, MNIST, RESISC45, SUN397, and SVHN. Moving to 14 tasks, we add CIFAR100, STL10, Flowers102, OxfordIIITPet, PCAM, and FER2013. The 20-task suite further includes EMNIST, CIFAR10, Food101, FashionMNIST, RenderedSST2, and KMNIST. We provide the specific dataset details in Table 5. For our scalability study we further include 13 additional publicly available datasets to reach a total of 33 tasks. These additional datasets are collected form the EMR-Merging (Huang et al., 2024) thirty tasks benchmark: Beans, CUB200, Dogs, FlowersKaggle, Fruits360, Garbage, IntelImages, KenyanFood13, KvasirV2, Landscape, MangoLeafBD, Vegetables, and Weather.

---

[2]https://pytorch.org/vision/stable/index.html

| Dataset | image size | # train | # val | # test |
|---|---|---|---|---|
| Cars (Krause et al., 2013) | varies | 7330 | 814 | 8041 |
| DTD (Cimpoi et al., 2014) | varies | 1692 | 188 | 1880 |
| EuroSAT (Helber et al., 2019) | $64 \times 64$ | 21600 | 2700 | 2700 |
| GTSRB (Stallkamp et al., 2011) | varies | 23976 | 2664 | 12630 |
| MNIST (Lecun et al., 1998) | $28 \times 28$ | 55000 | 5000 | 10000 |
| RESISC45 (Cheng et al., 2017) | $256 \times 256$ | 17010 | 1890 | 6300 |
| SUN397 (Xiao et al., 2016) | varies | 17865 | 1985 | 19850 |
| SVHN (Netzer et al., 2011) | $32 \times 32$ | 68257 | 5000 | 26032 |
| CIFAR100 (Krizhevsky & Hinton, 2009) | $32 \times 32$ | 45000 | 5000 | 10000 |
| STL10 (Coates et al., 2011) | $96 \times 96$ | 4500 | 500 | 8000 |
| Flowers102 (Nilsback & Zisserman, 2008) | varies | 918 | 102 | 6149 |
| OxfordIIITPet (Parkhi et al., 2012) | varies | 3312 | 368 | 3669 |
| PCAM (Veeling et al., 2018) | $96 \times 96$ | 257144 | 5000 | 32768 |
| FER2013 (Goodfellow et al., 2013) | $48 \times 48$ | 25839 | 2870 | 7178 |
| EMNIST (Cohen et al., 2017) | $28 \times 28$ | 235000 | 5000 | 40000 |
| CIFAR10 (Krizhevsky & Hinton, 2009) | $32 \times 32$ | 45000 | 5000 | 10000 |
| Food101 (Bossard et al., 2014) | $512 \times 512$ | 70750 | 5000 | 25250 |
| FashionMNIST (Xiao et al., 2017) | $28 \times 28$ | 55000 | 5000 | 10000 |
| RenderedSST2 (Socher et al., 2013) | varies | 6228 | 692 | 1821 |
| KMNIST (Clanuwat et al., 2018) | $28 \times 28$ | 55000 | 5000 | 10000 |
| Beans (Lab, 2020) | $500 \times 500$ | 1030 | 133 | 128 |
| CUB200 (Wah et al., 2011) | varies | 5999 | 0 | 5790 |
| Dogs (Khosla et al., 2011) | varies | 12000 | 0 | 5880 |
| FlowersKaggle | varies | 4242 | 0 | 942 |
| Fruits360 (Oltean, 2017) | $100 \times 100$ | 118783 | 0 | 39613 |
| Garbage (CCHANG, 2018) | varies | 2359 | 0 | 0 |
| IntelImages | $150 \times 150$ | 14000 | 3000 | 7000 |
| KenyanFood13 (Wang et al., 2019) | varies | 8,174 | 0 | 0 |
| KvasirV2 (Pogorelov et al., 2017) | varies | 4000 | 0 | 0 |
| Landscape | varies | 1000 | 1500 | 500 |
| MangoLeafBD (Ahmed et al., 2023) | $240 \times 320$ | 4000 | 0 | 0 |
| Vegetables Ahmed et al. (2021) | varies | 14700 | 3150 | 3150 |
| Weather (Xiao, 2021) | $256 \times 256$ | 6877 | 0 | 0 |

Table 5: Image sizes and numbers of train, validation, and test samples for all datasets considered. For datasets lacking a validation or test split we used the $10\%$ of the training set.

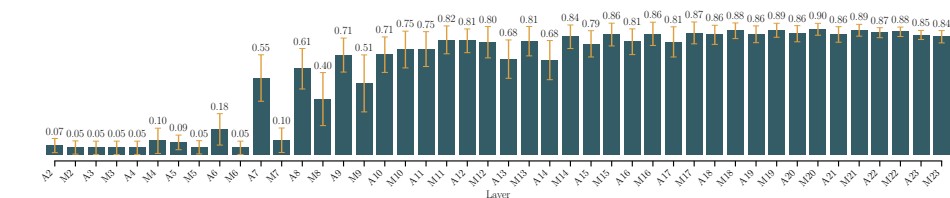

Figure 8: `ViT-L-14` per-layer task accuracies.

## B.4 EVALUATION MEASURES

To account for differences in task difficulty, we report both *absolute* and *normalized* accuracy in our results. The normalized accuracy serves as a relative performance measure by comparing the multi-task model's accuracy to that of individual fine-tuned models. It is computed as:

$$\text{Normalized Accuracy} = \frac{1}{T} \sum_{i=1}^{T} \frac{\text{accuracy}(\theta_{MT}, t_i)}{\text{accuracy}(\theta_{ft_i}, t_i)} \qquad (2)$$

where $T$ represents the total number of tasks, $\theta_{MT}$ is the multi-task model, and $\theta_{ft_i}$ corresponds to the fine-tuned model for task $t_i$. By normalizing accuracy in this way, we ensure a fairer comparison that accounts for variations in baseline task performance.

## B.5 ARCHITECTURES

We use CLIP from the *OpenClip* library[3], using the three different versions described in Table 6. For the router, we use a small two-layer MLP with a hidden dimension of 1024. It accepts a 512-dimensional embedding vector, applies a linear transformation, a ReLU activation, and dropout with a probability of 0.5, and outputs logits corresponding to task selection probabilities.

| Model | Layers | Hidden Dimension | Heads | Patch Size | Parameters |
|-------|--------|------------------|-------|------------|------------|
| `ViT-B-32` | 12 | 768 | 12 | 32×32 | ∼86M |
| `ViT-B-16` | 12 | 768 | 12 | 16×16 | ∼86M |
| `ViT-L-14` | 24 | 1024 | 16 | 14×14 | ∼307M |

Table 6: Comparison of `ViT-B-32`, `ViT-B-16`, and `ViT-L-14` architectures.

## B.6 ADAPTING ORACLE MOERGING METHODS

As extensively discussed, we argue that MoErging methods should not rely on an oracle head, since their pipelines implicitly assume that the task label is unknown at inference time (otherwise, merging the correct task vector would be trivial). To respect this assumption, we introduced a routing procedure that exploits the routers already implemented in each method. For SMILE, we extracted the mode of the tokens at each layer and applied a naïve majority-voting scheme across layers to select the head for each sample, which was then used for the final classification. Analogously, for WeMoE, we applied the same logic, implementing a straightforward head-selection strategy that leverages the existing gates and coefficients. All results reported for MoE methods were obtained under this unified setting.

**Radar charts on 8- and 14-task benchmarks.** In section Section 4, we present comprehensive results for the approach using the 20 task benchmark. Figure 9 and Figure 11 respectively display the normalized accuracies for our method on 8 and 14 tasks across all three model sizes (`ViT-B-32`, `ViT-B-16`, and `ViT-L-14`). In both cases, the approach retains a high fraction of each fine-tuned

---
[3]https://github.com/mlfoundations/open_clip/

---

**Algorithm 2** Fixed Merging Step

---

**Require:** Pretrained model weights $\theta_{\text{pre}}$, task-specific updates $\{\Delta_i\}_{i=1}^{T}$, user-specified threshold $\varepsilon$
**Ensure:** Fixed merged model weights $\theta_{\text{MT}}$
  1: **Accounting for redundant directions** (Section 3.2.2)
  2: $\mathcal{M} = \{\}$
  3: **for** $i = 1, \ldots, T$ **do**
  4:     $\delta_i \leftarrow \text{vec}(\Delta_i)$
  5:     **if** $\max_{\{j \in \mathcal{M}\}} \text{sim}(\delta_i, \delta_j) < \varepsilon$ **then**
  6:         $\mathcal{M} \leftarrow \mathcal{M} \cup \{i\}$
  7:     **end if**
  8: **end for**
  9: **Merging step using** `TSV-M` (Gargiulo et al., 2025) **on the** $\{\Delta_i\}_{i \in \mathcal{M}}$
 10: **for** $i \in \mathcal{M}$ **do**
 11:     $\Delta_i = U_i \Sigma_i V_i^\top$
 12:     $\tilde{U}_i \leftarrow U_{i[:,1:k]}, \tilde{\Sigma}_i \leftarrow \Sigma_{i[1:k,1:k]}, \tilde{V}_i \leftarrow V_{i[:,1:k]}$
 13: **end for**
 14:     $U \leftarrow [\tilde{U}_1 \,|\, \tilde{U}_2 \,|\, \cdots \,|\, \tilde{U}_T]$
 15:     $\Sigma \leftarrow \text{block\_diag}(\tilde{\Sigma}_1, \tilde{\Sigma}_2, \ldots, \tilde{\Sigma}_T)$
 16:     $V \leftarrow [\tilde{V}_1 \,|\, \tilde{V}_2 \,|\, \cdots \,|\, \tilde{V}_T]$
 17:     $U_\perp \leftarrow \text{orthogonalize}(U)$
 18:     $V_\perp \leftarrow \text{orthogonalize}(V)$
 19:     $\hat{\Delta} \leftarrow U_\perp \Sigma V_\perp^\top$
 20:     $\theta_{\text{MT}} \leftarrow \theta_{\text{pre}} + \alpha \, \hat{\Delta}$
 21: **return** $\theta_{\text{MT}}$

---

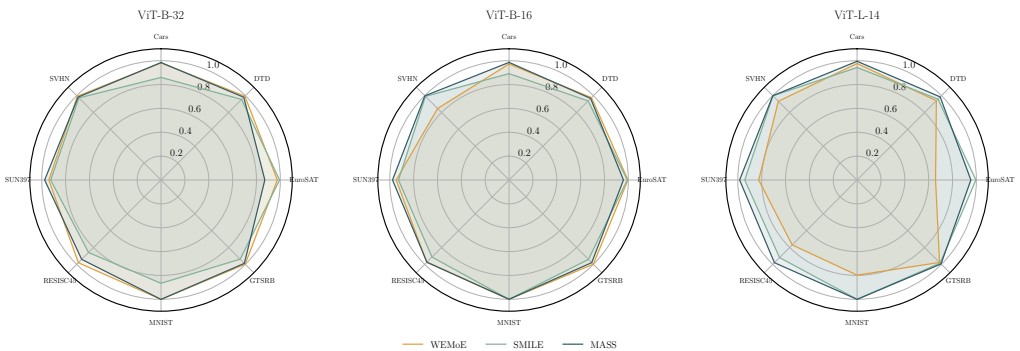

Figure 9: Normalized task accuracies over models `ViT-B-32`, `ViT-B-16` and `ViT-L-14` for the 8 tasks benchmark.

model's performance, with normalized accuracies often above 80–90%. Notably, the method scales gracefully as the number of tasks increases from 8 to 14.

## C    ADDITIONAL EXPERIMENTS AND RESULTS

In this section, we provide detailed per-task and per-layer accuracy plots, along with further examples of decoded task vectors, to complement the results presented in the main paper.

### C.1    MORE EXPERIMENTS

**Generalization to unseen tasks**    To evaluate how well the method behaves on tasks not included during merging, we conducted an out-of-distribution experiment on the 8-task benchmark. For each task in turn, we removed it from the merge set, merged the remaining seven tasks, and then evaluated on the held-out one. The Table 7 reports the results, comparing MASS with Task Arithmetic. `MASS`

outperforms the static merging baseline in all cases, this suggests that the task subspaces extracted from weight updates contain transferable structure that remains useful even when the target task was never seen during merging.

| Method | ViT-B-32 | | | | | | | |
|---|---|---|---|---|---|---|---|---|
| | DTD | MNIST | GTSRB | SVHN | EuroSAT | RESISC45 | Cars | SUN397 |
| TA | $31.5_{(40.3)}$ | $79.8_{(80.2)}$ | $26.7_{(27.0)}$ | $49.7_{(51.4)}$ | $31.0_{(31.3)}$ | $36.9_{(38.7)}$ | $40.6_{(50.9)}$ | $44.4_{(59.3)}$ |
| **MASS** | $\mathbf{36.9_{(47.2)}}$ | $\mathbf{86.6_{(87.1)}}$ | $\mathbf{33.1_{(33.4)}}$ | $\mathbf{59.7_{(61.8)}}$ | $\mathbf{40.9_{(41.2)}}$ | $\mathbf{46.4_{(48.6)}}$ | $\mathbf{49.5_{(62.0)}}$ | $\mathbf{54.8_{(73.2)}}$ |

Table 7: Comparison of `MASS` and `Task Arithmetic` on 8 datasets using `ViT-B-32`. Main accuracy with normalized accuracy in subscript.

**Scalability to Large Task Sets.** Table 8 summarizes our scalability evaluation presented in Fig. 6 providing exact numerical results. Across all configurations, MASS maintains a substantial performance margin over task arithmetic. Even at 33 tasks, our method retains a gap of roughly 20 accuracy points over the baseline. Importantly, the method preserves efficiency: the overall computational cost continues to be dominated by two forward passes, which do not depend on the number of tasks.

| Method | ViT-B-32 | | | |
|---|---|---|---|---|
| | $n=20$ | $n=25$ | $n=30$ | $n=33$ |
| TA | 0.72 | 0.63 | 0.64 | 0.64 |
| **MASS** | **0.91** | **0.85** | **0.83** | **0.83** |

Table 8: Scaling MASS to larger task sets.

The only term that grows with the number of tasks is the routing computation, each step projects intermediate activations onto the task subspaces via a small matrix product, resulting in routing complexity that scales linearly with the number of tasks. This ensures that MASS continues to merge large task collections without degradation in compute efficiency.

**Weighted vs Optimal TSV.** Prior work (Gargiulo et al., 2025) has established that the optimal $\alpha$ for `TSV` is 1.0 and remains invariant across a range of task counts (8–20 tasks). We implemented a variant of `TSV` in which the left and right singular vectors are scaled by $w = \text{softmax}(-r)$, before the merging with using residuals norm as a measure of contribution. The results indicate that the weighted `TSV` does not improve performance compared to our original approach, confirming the optimality of the value established in prior work.

| Method | 8 tasks | 14 tasks | 20 tasks |
|---|---|---|---|
| Weighted | 93.5 | 90.1 | 86.4 |
| Ours | 96.5 | 93.2 | 90.9 |

Table 9: Comparison of weighted `TSV` and classic `TSV` on `ViT-B-32` across different task counts.

**Adaptive Merging Ablation.** We investigated the impact of singular vectors orthogonalization in our adaptive merging step in Table 10. The results show that applying orthogonalization restricted to the routed set does provide a consistent benefit, but the gains are modest. This is expected: for small task sets, `Task Arithmetic` and `TSV` already perform similarly, since interference between tasks is limited and the dominant singular directions are relatively well separated. Orthogonalization becomes more useful as the number of tasks increases and subspace overlap grows, but it is not the primary driver of the method's performance.

| Method | ViT-B-32 | | | ViT-B-16 | | | ViT-L-14 | | |
|---|---|---|---|---|---|---|---|---|---|
| | 8 tasks | 14 tasks | 20 tasks | 8 tasks | 14 tasks | 20 tasks | 8 tasks | 14 tasks | 20 tasks |
| non-ortho | 95.3 | 93.0 | 90.7 | 97.3 | 96.1 | 88.5 | 97.8 | 97.3 | 96.6 |
| ortho | 96.5 | 93.2 | 90.9 | 98.0 | 96.1 | 88.7 | 98.6 | 97.3 | 96.6 |

Table 10: Effect of orthogonalization in adaptive merging. Applying orthogonalization restricted to the routed set $\Omega$ consistently improves performance across `ViT-B-32`, `ViT-B-16`, and `ViT-L-14`.

| Method | ViT-B-32 | | | ViT-B-16 | | |
|---|---|---|---|---|---|---|
| | 8 tasks | 14 tasks | 20 tasks | 8 tasks | 14 tasks | 20 tasks |
| **SMILE (oracle head)** | 87.7 | 85.9 | 85.0 | 92.8 | 92.0 | 91.4 |
| **MASS (oracle head)** | **89.3** | **86.6** | **86.2** | **93.7** | **92.8** | **91.5** |

Table 11: Comparison of MASS and SMILE under the oracle head setting across ViT-B-32 and ViT-B-16 for different task counts.

**Oracle Head**   The novel experimental setting introduced in this paper considers the classification head not known a priori, requiring a router to select this one alongside the optimal encoder. To ensure a comprehensive evaluation, we however also compared our approach against the best performing baseline SMILE also under the classical oracle head setting. Table 11 shows the results. While the gap closes, MASS still outperforms SMILE across all benchmarks proving again its effectiveness.

## C.2   HYPERPARAMETER SETTINGS

Following the recommendation of TSV (Gargiulo et al., 2025), we use $\alpha$ as a single scaling factor with the suggested value of 1.0 for the TSV-M merging configurations in both Algorithm 2 and Algorithm 1. Consistent with TSV, the compression rate assigned to each task space is set to $\frac{1}{T}$. We optimized the similarity threshold $\varepsilon$ over the range {0.1, 0.2, ..., 0.9} and determined the router's selection threshold $\eta$ via a Bayesian search within the interval [0.05, 0.5]. As illustrated in Figures 4a and 8, we identify the optimal projection layer by focusing on those revealing the highest task accuracy. Specifically, for ViT-B-32 and ViT-B-16 models, we select the attention and MLP layers within the range {7, ..., 11}, while for the ViT-L-14 model, the chosen layers fall in the range {19, ..., 23}. The temperature parameter for tuning the behavior of the softmax function at line 6 in Algorithm 1 is set to 1.

**Sensitivity to Hyperparameters**   The merging coefficient is set to 1 as in TSV-M (Gargiulo et al., 2025). Figure 10 shows how accuracy changes with routing threshold $\eta$ and top-$K$. At low $\eta$ (0.05) and large $K$, too many tasks are merged, causing interference. At high $\eta$ ($\geq 0.3$), only the top task is selected, making performance insensitive to $K$ (it's effectively an argmax). The best accuracy occurs in a broad middle range, peaking at $\eta = 0.2$, $K = 3$, where the router balances selectivity and coverage. We also vary the cosine threshold $\varepsilon$ used to discard similar task updates before merging. Due to the high dimensionality of the $\Delta$s, large thresholds ($\varepsilon \geq 0.4$) retain all updates, leaving redundancy unaddressed (accuracy 93.5). Small ones ($\varepsilon \leq 0.05$) instead remove even distinct directions, significantly harming accuracy ($\leq 88.6$). Intermediate values ($\varepsilon \approx 0.2$) offer a robust filtering, improving performance ($\geq 93.9$) by suppressing redundancy.

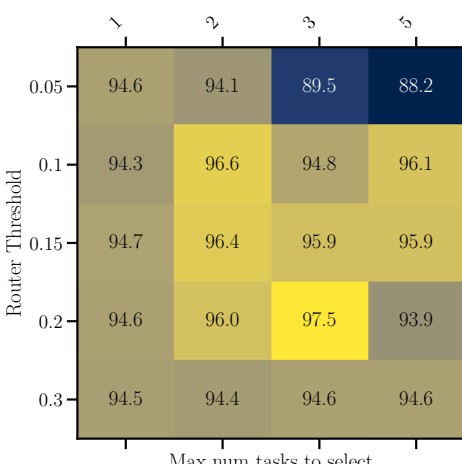

Figure 10: Hyperparameter sensitivity.

**Layer-wise accuracies for individual datasets.**   Figures 12 and 13 show per-layer accuracies for ViT-B-32 on different subsets of the 8-task benchmark:

- Figure 12 focuses on Cars, DTD, EuroSAT, and GTSRB.

- Figure 13 displays results for MNIST, RESISC45, SUN397, and SVHN.

Again, the top-performing layer is not shared across the tasks, confirming what we observed in the main paper.

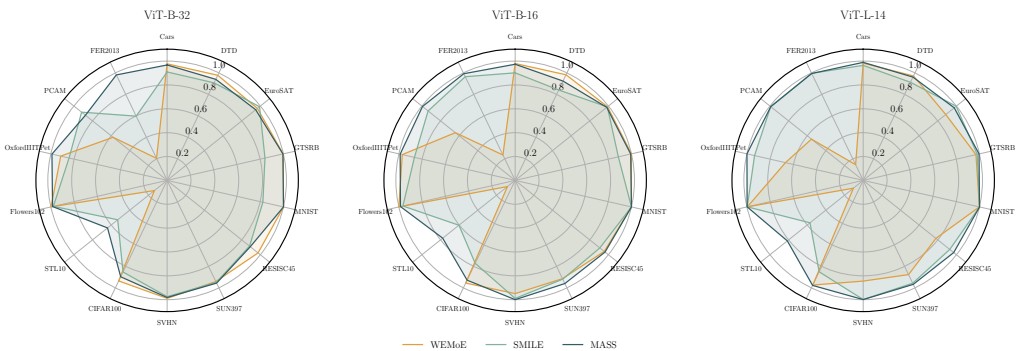

Figure 11: Normalized task accuracies over models `ViT-B-32`, `ViT-B-16` and `ViT-L-14` for the 14 tasks benchmark.

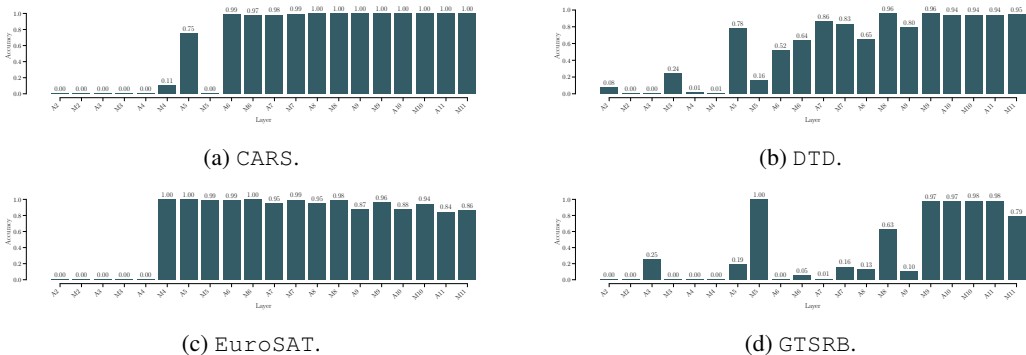

Figure 12: Per-layer task accuracies for `ViT-B-32` on `Cars`, `DTD`, `EuroSAT`, and `GTSRB`.

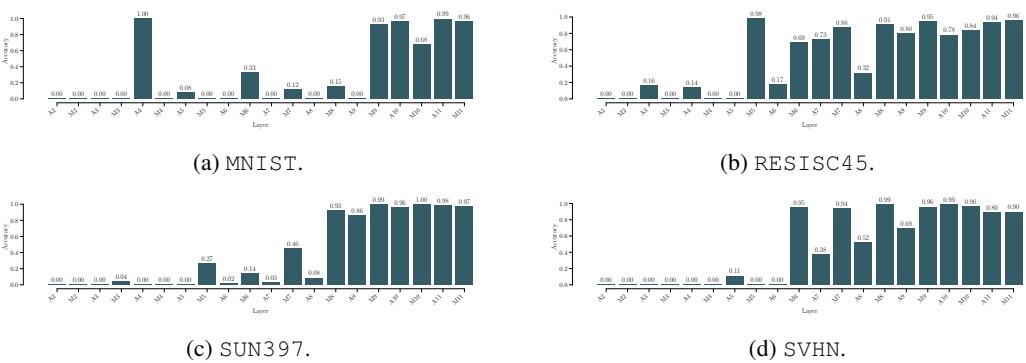

Figure 13: Per-layer task accuracies for `ViT-B-32` on `MNIST`, `RESISC45`, `SUN397`, and `SVHN`.

**Layer-wise accuracies for `ViT-L-14`.** Figure 8 reports average per-layer task accuracy for the larger `ViT-L-14`, showing that in this case the most predictive layer for routing is $\ell = 20$. Since `ViT-L-14` has 24 layers (compared to the 12 layers in `ViT-B-32` and `ViT-B-16`), the most predictive layer is roughly at the same relative depth.

**Visualizations for additional datasets.** Following the approach in Section 4.4, Figure 14 shows examples of decoded task vectors for datasets like `SVHN`, `GTSRB`, `SUN397`, and `RESISC45`. Here, we see textual prompts such as "*An image of the number 4*" or "*Aerial view of an industrial area*", which align with each dataset's distinct domain. This reaffirms that our singular vectors capture domain-specific transformations while preserving high-level semantic alignment to the pretrained model.

**Replacing $L_2$ with Mahalanobis distance.** Alternatively to the $L_2$-based projection used for routing, one could employ a data-free Mahalanobis distance by whitening the task-specific bases derived from the right singular vectors used for projection. Being all statistics computed solely from the learned bases, this would still preserve the data-free assumption.

| | ViT-B-32 | | |
|---|---|---|---|
| | 8 tasks | 14 tasks | 20 tasks |
| **Mahalanobis** | 95.7 | 91.0 | 88.0 |
| **L2 (Ours)** | **96.5** | **93.2** | **90.9** |

Table 12: Comparison of Mahalanobis vs. L2 distance.

In particular, given the right singular vectors $(V_i \in \mathbb{R}^{R \times D})_{(i=0,...,T)}$, with $R$ chosen such that $R \times T = D$, we concatenate and center them: $V_c = V - \mu_V$, $\mu_V \in \mathbb{R}^R$, and compute the covariance: $\Sigma = \mathrm{cov}(V_c) \in \mathbb{R}^{D \times D}$. The Mahalanobis distance between $x$ and $y$ is then:

$$d(x, y) = \sqrt{(x - y)^\top \Sigma^{-1} (x - y)}.$$

We report the corresponding results in Table 13. Whitening the bases and replacing $L_2$ with Mahalanobis does not improve performance; results are consistently slightly worse, indicating that the Euclidean residual is sufficiently well-conditioned for our routing mechanism.

**Cosine similarity vs principal angles for redundancy filtering** We implemented the proposed approach by computing the principal angles between subspaces (PABS (Åke Björck & Golub, 1973)), pairs of task matrices $\Delta_i$. For each candidate $\Delta_i$, we used the largest singular value (equivalently, the minimum cosine between the subspaces) as the similarity measure when deciding whether the update was already "represented" by previously accepted ones. As with the cosine-based criterion, we performed a small grid search over the threshold on the 8-task benchmark. We report the normalized accuracies below.

| | ViT-B-32 | | |
|---|---|---|---|
| | 8 tasks | 14 tasks | 20 tasks |
| **PBAS** | **96.8** | 92.7 | 86.8 |
| **Cosine** | 96.5 | **93.2** | **90.9** |

Table 13: Comparison of Principal Angles Between Subspaces vs. Cosine.

The principal-angle criterion slightly outperforms cosine similarity on the 8-task benchmark, but loses accuracy on the larger 14- and 20-task settings. This suggests that while more geometrically principled, the principal-angle filtering is less robust when task diversity increases.

**Per-task variances** We provide in table Table 14 the task-wise mean and standard deviation for the MoErging experiments. Remarkably, MASS not only has a better average, but also a smaller standard deviation with respect to the existing baselines.

## D    PROPOSITIONS AND PROOFS

**Proposition D.1** (Optimality of Orthogonal Projection). *Let $V \in \mathbb{R}^{d \times k}$ have orthonormal columns spanning a subspace $\mathcal{S} \subseteq \mathbb{R}^d$, and let $\mathbf{a} \in \mathbb{R}^d$. Then the unique minimizer of $\|\mathbf{a} - \mathbf{w}\|_2^2$ over all $\mathbf{w} \in \mathcal{S}$ is*

$$\widehat{\mathbf{w}} = V V^\top \mathbf{a}.$$

| Method | ViT-B-32 | | | ViT-B-16 | | | ViT-L-14 | | |
| | 8 tasks | 14 tasks | 20 tasks | 8 tasks | 14 tasks | 20 tasks | 8 tasks | 14 tasks | 20 tasks |
|---|---|---|---|---|---|---|---|---|---|
| WeMoE | $97.74 \pm 2.19$ | $82.83 \pm 28.78$ | $76.36 \pm 33.45$ | $96.43 \pm 4.72$ | $83.26 \pm 28.99$ | $70.54 \pm 32.51$ | $94.29 \pm 7.50$ | $72.51 \pm 35.64$ | $61.41 \pm 32.91$ |
| SMILE-1 | $92.14 \pm 5.02$ | $84.56 \pm 12.63$ | $82.32 \pm 20.31$ | $94.92 \pm 3.78$ | $89.56 \pm 10.18$ | $86.75 \pm 15.29$ | $96.78 \pm 2.90$ | $92.78 \pm 10.73$ | $90.52 \pm 13.72$ |
| SMILE-2 | $93.58 \pm 4.67$ | $85.70 \pm 13.11$ | $83.86 \pm 20.56$ | $96.20 \pm 2.83$ | $90.70 \pm 10.42$ | $87.77 \pm 15.33$ | $97.63 \pm 2.38$ | $93.42 \pm 10.93$ | $91.15 \pm 13.93$ |
| **MASS** | $\mathbf{96.54 \pm 3.94}$ | $\mathbf{93.23 \pm 9.02}$ | $\mathbf{90.90 \pm 9.61}$ | $\mathbf{98.01 \pm 1.33}$ | $\mathbf{96.17 \pm 5.63}$ | $\mathbf{88.74 \pm 19.89}$ | $\mathbf{98.68 \pm 1.43}$ | $\mathbf{97.39 \pm 4.63}$ | $\mathbf{96.65 \pm 4.25}$ |

Table 14: Average performance (mean $\pm$ std) on 8-, 14-, and 20-task benchmarks for different ViT backbones.

| MASS | ViT-B-32 | | | ViT-B-16 | | |
| + | 8 tasks | 14 tasks | 20 tasks | 8 tasks | 14 tasks | 20 tasks |
|---|---|---|---|---|---|---|
| nn | 92.7 | 89.6 | 89.4 | 92.7 | 90.0 | 90.2 |
| mlp | 96.8 | 95.8 | 95.8 | 97.1 | 94.8 | 96.5 |
| $\texttt{proj}_{\text{PRE}}$ | 98.2 | 88.6 | 79.4 | 98.7 | 92.0 | 81.2 |
| $\texttt{proj}_{\text{TSV-M}}$ | 96.5 | 93.2 | 90.9 | 98.0 | 96.1 | 88.7 |

Table 15: Average normalized accuracy for different routers.

*Proof.* Any $\mathbf{w} \in \mathcal{S}$ can be written as $V\,\boldsymbol{\alpha}$ for some $\boldsymbol{\alpha} \in \mathbb{R}^k$. The problem

$$\min_{\mathbf{w} \in \mathcal{S}} \|\mathbf{a} - \mathbf{w}\|_2^2 \quad \Longleftrightarrow \quad \min_{\boldsymbol{\alpha} \in \mathbb{R}^k} \|\mathbf{a} - V\,\boldsymbol{\alpha}\|_2^2$$

has a strictly convex objective, so its global minimizer is found by setting the gradient to zero. A short calculation shows

$$\boldsymbol{\alpha} = V^\top \mathbf{a} \quad \Longrightarrow \quad \widehat{\mathbf{w}} = V(V^\top \mathbf{a}) = V\,V^\top\,\mathbf{a}.$$

Uniqueness follows from the strict convexity, and $\|\mathbf{a} - \widehat{\mathbf{w}}\|_2$ is necessarily the smallest possible distance in $\mathcal{S}$. Equivalently, $\mathbf{a} - \widehat{\mathbf{w}}$ is orthogonal to $\mathcal{S}$, so no further reduction in norm is possible. $\square$

**Proposition D.2** (§3.1). *Let $z_\ell \in \mathbb{R}^d$ be a feature vector, and for each task $i$, decompose it as*

$$z_\ell = V_i V_i^\top z_\ell + \varepsilon_i, \qquad \varepsilon_i = \left(I - V_i V_i^\top\right) z_\ell.$$

*Assume $\varepsilon_i \sim \mathcal{N}(0, \sigma^2 I)$. Then the maximum a posteriori estimate of the task reduces to*

$$\hat{i}_{\text{MAP}} = \arg\max_i p(task = i \mid z_\ell) = \arg\min_i \|\varepsilon_i\|_2^2.$$

*Thus, under these assumptions, selecting the task with the smallest squared Euclidean residual is exactly equivalent to maximizing the posterior.*

*Proof.* By assumption, $p(\varepsilon_i) = (2\pi\sigma^2)^{-d/2} \exp\!\left(-\|\varepsilon_i\|_2^2/(2\sigma^2)\right)$, so $-\log p(\varepsilon_i) \propto \|\varepsilon_i\|_2^2$. Since $V_i V_i^\top z_\ell$ is a deterministic shift, the likelihood $p(z_\ell \mid task = i)$ depends only on $\varepsilon_i$. With a uniform prior over tasks,

$$\hat{i}_{\text{MAP}} = \arg\max_i p(z_\ell \mid task = i) = \arg\max_i p(\varepsilon_i) = \arg\min_i \|\varepsilon_i\|_2^2.$$

Hence, minimizing the $\ell_2$ residual is exactly equivalent to maximizing the posterior. $\square$

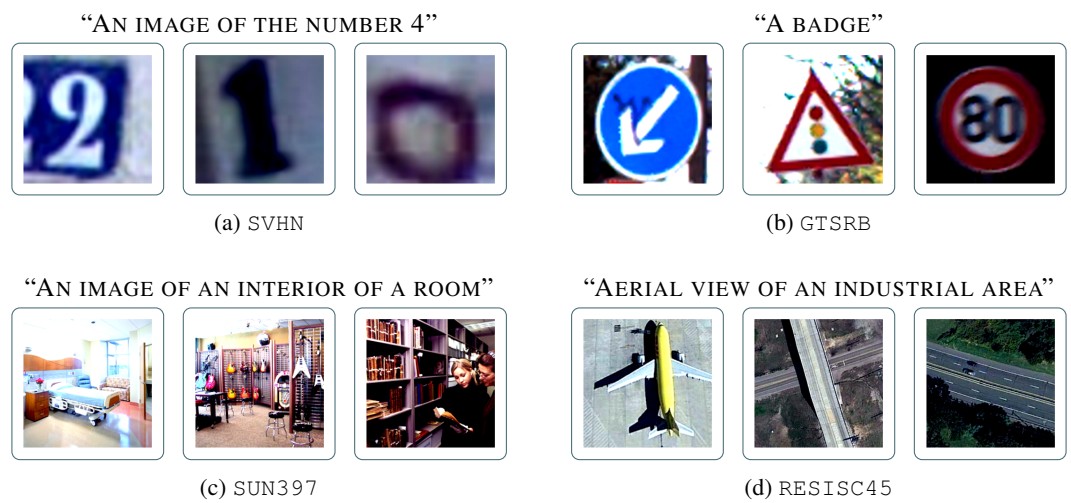

"AN IMAGE OF THE NUMBER 4"

(a) SVHN

"A BADGE"

(b) GTSRB

"AN IMAGE OF AN INTERIOR OF A ROOM"

(c) SUN397

"AERIAL VIEW OF AN INDUSTRIAL AREA"

(d) RESISC45

Figure 14: Captions obtained by decoding task singular vectors as text for datasets SVHN, GTSRB, SUN397, and RESISC45 as described in Section 4.4, accompanied by three representative images for each dataset.

