# OpenReview forum: "MASS: MoErging through Adaptive Subspace Selection"
_ICLR.cc/2026/Conference — ICLR 2026 Poster_

### Official Review · Reviewer_JZ1o · 2025-10-25

**Soundness:** 3
**Presentation:** 3
**Contribution:** 2
**Rating:** 4
**Confidence:** 4

**Summary:**

This paper introduces MASS (MoErging through Adaptive Subspace Selection), a novel model merging approach, and its core premise is that in practical applications, the task identity is often unknown at inference time. To tackle this more challenging and realistic scenario, MASS introduces an adaptive routing mechanism. This mechanism, which is free of additional training and data, dynamically selects the most relevant task subspace for each input sample by calculating the projection residuals of the input's features onto different task subspaces. Experimental results demonstrate that MASS achieves state-of-the-art performance on several benchmarks.

**Strengths:**

1.	A Critical and Practical Problem Setting: The paper astutely identifies and challenges an unrealistic assumption common in existing model merging literature: that task identity is known at inference time. By proposing a more realistic scenario where task identity is unknown, the authors elevate the practical relevance of their research and open up a more meaningful direction for the community. MASS provides a complete and effective end-to-end solution for this setting.
2.	A Novel and Efficient Training-Free Routing Mechanism: A major highlight of this work is its projection-based router. This process is entirely free of additional labeled data and training overhead, which aligns perfectly with the lightweight and training-free philosophy of model merging. And it performs on par with MLP-based routers that require additional data and training.

**Weaknesses:**

1.	Limited Contribution: The paper's core framework is heavily built upon prior work, particularly Task Singular Vectors (TSV). Consequently, the core contribution of this paper can be viewed as the addition of an efficient, dynamic "selector" module on top of the TSV framework. While this selector is well-designed and effective, it functions more as a task scheduling or selection component rather than a fundamentally paradigm for “adaptive”  model merging. This makes the overall contribution of the paper feel somewhat incremental.
2.	Insufficient Experimental Scope: The paper's experiments are primarily focused on ViT-based vision classification tasks and NLU tasks. However, a more challenging and highly relevant application for model merging is on large language models (LLMs) performing difficult generative tasks (e.g., code generation, long-form text generation). For such tasks, task vectors often exhibit more complex, high-rank structures. Previous research[1, 2, 3] has questioned that simply relying on low-rank decomposition may not perform optimally when dealing with these high-rank task vectors. Given that MASS's performance is highly dependent on the effectiveness of TSV, its performance on LLMs and generative tasks remains a critical, unverified question. The absence of these experiments raises questions about the generalizability of MASS and its effectiveness on cutting-edge applications.
[1] Delta-CoMe: Training-Free Delta-Compression with Mixed-Precision for Large Language Models
[2] Knowledge Grafting of Large Language Models
[3] Delta Decompression for MoE-based LLMs Compression

**Questions:**

See weaknesses

---

> ### Author Response · Authors · 2025-11-21
>
> We thank the reviewer for their feedback. We are glad they appreciated the novel training-free routing mechanism and praised the practicality of the setting. We address their concerns below.
>
> **W1 — Incrementality of the contribution.**
>
> We respectfully disagree with the characterization that the contribution is merely incremental. In MoErging, the routing mechanism is not an auxiliary component but the core element that determines whether adaptive merging is possible at all. A router that is trained with task data, as in TwinMerging (NeurIPS’24) [1], is relatively straightforward to construct because it effectively learns to classify tasks from labelled examples. **In our setting, however, neither data nor training is available,** making the design of a *data-free*, *training-free* router that operates solely from the weight updates substantially more challenging.
>
> Furthermore, almost all MoErging works reuse existing expert-aggregation schemes [2], and their primary innovations lie precisely in how experts are selected, weighted, or dynamically combined. In this context, our projection-based routing built directly from the singular subspaces of the fine-tuned models enables ***fully adaptive* merging without oracle task labels, without data, and without learned gating.** This moves beyond a static extension of TSV and provides a plug-and-play adaptive mechanism that can be used on **any model**, even if downloaded from HuggingFace without any access to its data.
>
> Taken together, the data-free routing, automatic head selection, and dynamic subspace activation constitute the central contribution of the paper and, we believe, a meaningful step forward for effective and adaptive merging methods.
>
> [1] Lu, Zhenyi, et al. "Twin-merging: Dynamic integration of modular expertise in model merging." Advances in Neural Information Processing Systems 37 (2024): 78905-78935.
>
> [2] Yadav, Prateek, et al. "A Survey on Model MoErging: Recycling and Routing Among Specialized Experts for Collaborative Learning." Transactions on Machine Learning Research.

---

> ### Author Response · Authors · 2025-11-21
>
> **W2 — Insufficient experimental scope.**
>
> We agree that extending the study to decoder-only LLMs is a compelling direction for future work. At the same time, we emphasize that this extension is non-trivial. Merging open-vocabulary vision models or encoder-decoder/encoder-only language models is centered around *clear task boundaries*: each fine-tuned checkpoint corresponds to a distinct dataset with a well-defined label space, and the performance of the fine-tuned model provides a practical upper bound.
>
> In contrast, merging decoder-only LLMs often aims to create *new capabilities that none of the experts individually possess*. This involves skill composition, cross-domain reasoning, code synthesis, or long-form generation, settings where tasks are not discrete datasets but broad behaviors. As a result, the research lines for the two domains have diverged. Methods such as TIES [14] and DARE [15] are strong baselines for LLM merging, but they are outdated or suboptimal in vision. Conversely, structured approaches such as TSV [3], Iso-C [7], and KNoTS [12] provide clear advantages in vision/encoder models but do not yield gains when applied to decoder-only LLMs, where task vectors typically have high intrinsic rank and different interference patterns.
>
> Our work sits squarely in the first line of research, adaptive merging of models with explicit task boundaries, **where preserving each expert’s accuracy is the main goal**, and where low-rank structure has repeatedly been shown to be effective. This is the setting studied in a large body of prior work (a non-exhaustive list is provided in references [1–13]), all published at major conferences and centered around encoder-based architectures.
>
> We see our contribution as complementary to the LLM-merging literature rather than competing with it. We expect future work to adapt or rethink the routing mechanism introduced here for more compositional, high-rank regimes, and we view that as an exciting research direction.
>
> [1] Ilharco, Gabriel, et al. "Editing models with task arithmetic." The Eleventh International Conference on Learning Representations.
>
> [2] Wang, Ke, et al. "Localizing task information for improved model merging and compression." Proceedings of the 41st International Conference on Machine Learning. 2024.
>
> [3] Gargiulo, Antonio Andrea, et al. "Task singular vectors: Reducing task interference in model merging." Proceedings of the Computer Vision and Pattern Recognition Conference. 2025.
>
> [4] Lu, Zhenyi, et al. "Twin-merging: Dynamic integration of modular expertise in model merging." Advances in Neural Information Processing Systems 37 (2024)
>
> [5] Davari, MohammadReza, and Eugene Belilovsky. "Model breadcrumbs: Scaling multi-task model merging with sparse masks." *European Conference on Computer Vision*.
>
> [6] Ortiz-Jimenez, Guillermo, Alessandro Favero, and Pascal Frossard. "Task arithmetic in the tangent space: Improved editing of pre-trained models." Advances in Neural Information Processing Systems 36 (2023)
>
> [7] Marczak, Daniel, et al. "No Task Left Behind: Isotropic Model Merging with Common and Task-Specific Subspaces." Forty-second International Conference on Machine Learning.
>
> [8] Panariello, Aniello, et al. "Accurate and Efficient Low-Rank Model Merging in Core Space." *The Thirty-ninth Annual Conference on Neural Information Processing Systems*.
>
> [9] Yang, Enneng, et al. "AdaMerging: Adaptive Model Merging for Multi-Task Learning." The Twelfth International Conference on Learning Representations.
>
> [10] Daheim, Nico, et al. "Model Merging by Uncertainty-Based Gradient Matching." *ICLR 2024*. 2023.
>
> [11] Tam, Derek, Mohit Bansal, and Colin Raffel. "Merging by Matching Models in Task Parameter Subspaces." Transactions on Machine Learning Research.
>
> [12] Stoica, George, et al. "Model merging with SVD to tie the Knots." The Thirteenth International Conference on Learning Representations.
>
> [13] Tang, Anke, et al. "Merging multi-task models via weight-ensembling mixture of experts." *Proceedings of the 41st International Conference on Machine Learning*. 2024.
>
> [14] Yadav, Prateek, et al. "Ties-merging: Resolving interference when merging models." Advances in Neural Information Processing Systems 36 (2023): 7093-7115.
>
> [15] Yu, Le, et al. "Language models are super mario: Absorbing abilities from homologous models as a free lunch." *Forty-first International Conference on Machine Learning*. 2024.

---

> > ### Comment · Reviewer_JZ1o · 2025-11-26
> > **Thank the authors for the response in the rebuttal**
> >
> > Thank the authors for the response in the rebuttal.  I have carefully read the authors' rebuttal and plan to maintain my original score.

---

> > > ### Author Response · Authors · 2025-12-01
> > >
> > > We thank the reviewer for the update and for taking the time to re-read our response.
> > > However, we are disheartened by the dismissiveness of the rebuttal.
> > >
> > > Calling the work incremental because it leverages an existing merging method overlooks how virtually all MoErging methods build on prior aggregation schemes. The core novelty in this area lies in routing and selection. Dismissing a data-free, training-free router and the introduction of a practical generalist setting as incremental is not supported by how the field actually operates.
> > >
> > > Regarding the absence of large decoder-only LLM experiments, we reiterate that our work follows the established line of research focused on models with clear task boundaries. This includes merging open-vocabulary vision models and encoder-decoder or encoder-only language models. Decoder-only LLM merging is a different paradigm aimed at compositional skill acquisition, not task preservation. Declaring our evaluation setting insufficient directly contradicts the evaluation protocol used by more than a dozen prior works published at the top conferences in the field, all operating within the same regime and all using benchmarks that are strict subsets of ours.
> > >
> > > To fully address the only actionable concern, we additionally merged three Mistral-7B finetunings specialized in math and code: `meta-math/MetaMath-Mistral-7B`, `cognitivecomputations/dolphin-2.1-mistral-7b`, and `uukuguy/speechless-code-mistral-7b-v1.0`.
> > > Results:
> > >
> > > | Method  | MMLU | TruthfulQA | MathQA | ARC  |
> > > | ------- | ---- | ---------- | ------ | ---- |
> > > | Expert1 | 58.5 | 38.5       | 30.8   | 47.6 |
> > > | Expert2 | 61.0 | 46.9       | 37.4   | 55.9 |
> > > | Expert3 | 61.1 | 42.8       | 37.5   | 53.4 |
> > > | TA      | **61.6** | 43.3       | 37.0   | 54.1 |
> > > | MASS    | 61.4 | **47.4**       | **38.3**   | **55.7** |
> > >
> > > MASS outperforms all experts and clearly improves over standard Task Arithmetic on three out of four benchmarks, while remaining comparable on the last one. This preliminary experiment was the only one feasible within the rebuttal window, but it already shows clear promise that the approach extends to LLMs.

---

### Official Review · Reviewer_eoaD · 2025-10-30

**Soundness:** 2
**Presentation:** 3
**Contribution:** 2
**Rating:** 4
**Confidence:** 3

**Summary:**

This paper proposes applying model merging in a more challenging scenario, where the optimal encoder subspace and classification head for the current task are unknown. Under this setting, the authors introduce MASS, which predicts by selecting the most relevant series of encoders within the subspace. The method’s superiority over baselines is validated across tasks of varying scales.

**Strengths:**

- The authors provide detailed experimental results for analysis.
- The MASS method is simple yet effective.

**Weaknesses:**

- The description in Section 3.2.1 is somewhat unclear. Since minimizing $r_i$ yields the optimal $L_2$ projector, the proj in L191–192 should not be referred to as “the optimal $L_2$ projector.” Additionally, in line 195, what does “the residual” refer to—is it $r_i$?
- My main concern lies in the reasonableness of the paper’s setting.
    - Given that models have already been fine-tuned on each task, why is it necessary to assume that the task-optimal encoder and classification head are unknown? It is explicitly known which task each model has been fine-tuned on.
    - The authors should impose stricter constraints on the models involved in merging to make the setting reasonable.
    - Otherwise, the authors should clarify this point; otherwise, the foundation of the paper is difficult to justify.
- Line 406 mentions “prior work suggests that mid-layer embeddings in CLIP-like models...”; a citation should be added here.

**Questions:**

Please revise or clarify the description in Section 3.2.1.

---

> ### Author Response · Authors · 2025-11-16
>
> We thank the reviewer for their thoughtful feedback. We are glad you found the approach simple yet effective and the experimental analysis convincing. We address your concerns below.
>
> **W1 —** **Section 3.2.1 is unclear**. Thank you for pointing this out. We have revised the manuscript as follows:
>
> > “(...) where $Proj_{V_{i}^{(\ell)}} (z_{\ell}) = V_i^{(\ell)} V_i^{(\ell)^{\top}} z_{\ell}$ is the orthogonal projection of $z_\ell$ onto $\mathrm{span}(V_i^{(\ell)})$, which yields the minimum-error ($L_2$-optimal) reconstruction within that subspace.”
>
> Regarding the residuals, these are indeed the $r_i$s for each task $i = 1, \dots, T$. We revised the paper for clarity as follows:
>
> > At this point, the additive inverse of the residual vector $\mathbf{r} = (r_1, \dots , r_T)$ is normalized through a softmax to obtain the coefficients.
> >
>
> **W2 — Reasonableness of the setting.** We agree that the training task of each fine-tuned checkpoint is known. Our assumption however concerns the **task identity** of the input at inference time: when a merged model serves mixed, unlabeled traffic, the task of each *input sample* is not provided. In this regime, which was also praised by Reviewer JZ1o as **realistic and practical**, the model must infer for each input which encoder subspace and classification head to use. If the task were given at test time (oracle-task), the problem reduces to compression; TSV-C [1] already achieves ≥99.5% normalized accuracy with a slight extra storage requirement, effectively solving that setting. Our contribution targets the non-oracle, realistic case where task labels are absent at inference. This setup is aligned with the evaluation protocol introduced by KNoTS [2], designed to test generalist merging methods. The main difference is that we keep the individual classification heads available, since our routing mechanism can leverage them once the appropriate subspace is selected.
>
> In other words, prior model-merging methods typically assume that the task and corresponding label space of each input are known at inference. In our setting, this information is inferred automatically from the input itself, substantially broadening the method’s applicability in real deployments.
>
> **Missing citation in line 406.** Thank you for noting this. We revised “mid-layer” to “mid-to-late layer” (consistent with terminology already used in L423 and L428) and added a citation to [3], which explicitly states:
>
> > We find that ablating all layers but the last 4 attention layers has only a small effect on CLIP’s zero-shot classification accuracy (…) We conclude that the CLIP image representation is primarily constructed by these late attention layers.
> >
>
> We remain available for any further clarification.
>
> **References**
>
> [1] Gargiulo, Antonio Andrea, et al. "Task singular vectors: Reducing task interference in model merging." *Proceedings of the Computer Vision and Pattern Recognition Conference*. 2025.
>
> [2] Stoica, George, et al. "Model merging with SVD to tie the Knots." *The Thirteenth International Conference on Learning Representations*.
>
> [3] Gandelsman, Yossi, Alexei A. Efros, and Jacob Steinhardt. "Interpreting CLIP's Image Representation via Text-Based Decomposition." *The Twelfth International Conference on Learning Representations*.

---

### Official Review · Reviewer_8SY3 · 2025-10-31

**Soundness:** 2
**Presentation:** 2
**Contribution:** 3
**Rating:** 4
**Confidence:** 3

**Summary:**

The paper proposes a training‑ and data‑free way to merge multiple fine‑tuned models into a single network that selects relevant encoder subspaces per input and also selects the appropriate classification head when the task is unknown at test time. Concretely, each per‑task weight update $\Delta_i$ is SVD‑decomposed and truncated; at inference, an input’s mid‑layer features $z_\ell$ are projected onto each task’s right‑singular subspace and the residual $r_i=|z_\ell - V_i V_i^\top z_\ell|_2^2$ scores task relevance. The router keeps top‑K tasks above a softmax‑based threshold and adaptively merges the corresponding low‑rank updates; a second pass runs the selected heads and takes the max‑logit across them.

**Strengths:**

1. It Introduces a projection‑based, training‑free router in weight space that operates on TSV‑organized task subspaces and extends routing to head selection when the task is unknown. The MAP view provides a statistical rationale.
2. Algorithms and figures make the method easy to follow.
3. It Provides constant 2× storage independent of task count.

**Weaknesses:**

1. Routing layer and thresholds are selected with labeled validation accuracy as Appendix C. This partially contradicts the “data‑free” framing and may inflate reported gains versus truly label‑free procedures.
2. Alg. 1 uses $w=\mathrm{softmax}(-r)$ only for selection; the merge $\sum_{i\in\Omega}U_i\Sigma_iV_i^\top$ is uniform, which miss a straightforward weighted variant $\sum w_i\Delta_i$. An ablation could show whether weighting reduces interference or improves tail tasks.
3. The fixed merge (Alg. 2) uses orthogonalized bases $U_\perp,V_\perp$. It is unclear whether the adaptive merge applies the same orthogonalization restricted to $\Omega$. Clarify and ablate.
4. MoE baselines are adapted with a majority‑vote head heuristic. Provide companion results with oracle heads (as in their original setting) and/or head‑logit calibration (e.g., temperature scaling) to separate routing vs. head‑selection effects.
5. Euclidean residual assumes isotropy; unlabeled whitening/Mahalanobis distances on $z_\ell$ could be stronger and still data‑free, which relaxes Proposition 3.1’s assumptions. So it would be better to report sensitivity.
6. Cosine similarity of vec($\Delta$) ignores subspace geometry. Principal angles between spans $(V_i)$ would be more principled. Provide ablation.
7. No seeds/variance or CIs are given; wall‑clock overhead of two passes and multi‑head evaluation is not quantified; GLUE per‑task wins/losses are not detailed.

**Questions:**

See weakness.

---

> ### Author Response · Authors · 2025-11-21
>
> We thank the reviewer for their comments. We are glad they appreciated the training- and data-free router as well as the constant storage and the clarity of the manuscript. We address their concerns below.
>
> **W1 — Tuning of routing layers and thresholds.**
>
> We appreciate the concern. Although individual tasks may prefer different routing layers, the layers that offer the strongest *average* performance are consistently the deepest ones. In practice, this makes layer selection largely stable and not something that requires per-task or per-benchmark tuning.
>
> Regarding thresholds, we did tune them, but only once on the 8-task benchmark, and we then reused the exact same values for the 14- and 20-task settings. While it is possible that slightly better values exist for those larger benchmarks, the fact that the same thresholds work well across all settings suggests that reasonable choices generalize across task collections. This avoids any need for task-specific or benchmark-specific tuning and preserves the spirit of a data-free, training-free routing procedure. We will clarify this aspect in the revised manuscript.
>
> **W2 — Weighted variant.**
>
> Gargiulo et al. [1] have established that the optimal $\alpha$ for TSV is 1.0 and remains invariant across a range of task counts (8–20 tasks). To investigate the effect of weighting, we implemented a variant of TSV in which the left and right singular vectors are scaled by  $w = \text{softmax}(-r)$ , as suggested.
>
> | Method | B32 - 8 tasks  | B32 - 14 tasks | B32 - 20 tasks |
> | --- | --- | --- | --- |
> | weighted  | 93.5 | 90.1 | 86.4 |
> | optimal alpha | **96.5**	 | **93.2**	 | **90.9** |
>
> The results indicate that this weighted variant does not improve performance compared to our original approach, confirming the optimality of the $\alpha=1$ value established in prior work [1].
>
> **W3 — Orthogonalization in adaptive merging.**
>
> > *The fixed merge (Alg. 2) uses orthogonalized bases. It is unclear whether the adaptive merge applies the same orthogonalization restricted to $\Omega$. Clarify and ablate.*
> >
>
> We confirm that the adaptive merge applies the same orthogonalization procedure used in the fixed merge, restricted to the selected task set $\Omega$. Sharing the reviewer’s interest in the effect of this step, we ran an ablation in which the task singular vectors in $\Omega$ are *not* orthogonalized. Without orthogonalization, the procedure reduces to a low-rank variant of Task Arithmetic.
>
> The results below report normalized accuracies.
>
> | adaptive merging method | B32 – 8 tasks | B32 – 14 tasks | B32 – 20 tasks |
> | --- | --- | --- | --- |
> | non-ortho | 95.3 | 93.0 | 90.7 |
> | ortho | **96.5** | **93.2** | **90.9** |
>
> | adaptive merging method | B16 – 8 tasks | B16 – 14 tasks | B16 – 20 tasks |
> | --- | --- | --- | --- |
> | non-ortho | 97.3 | 96.1 | 88.5 |
> | ortho | **98.0** | **96.1** | **88.7** |
>
> | adaptive merging method | L14 – 8 tasks | L14 – 14 tasks | L14 – 20 tasks |
> | --- | --- | --- | --- |
> | non-ortho | 97.8 | 97.3 | 96.6 |
> | ortho | **98.6** | **97.3** | **96.6** |
>
> These results show that applying orthogonalization restricted to the routed set $\Omega$ does provide a consistent benefit, but the gains are modest. This is expected: for small task sets, Task Arithmetic and TSV already perform similarly, since interference between tasks is limited and the dominant singular directions are relatively well separated. Orthogonalization becomes more useful as the number of tasks increases and subspace overlap grows, but it is not the primary driver of the method’s performance. We will clarify the orthogonalization step and include these ablations in the revised manuscript.
>
> **W4 — Comparison with oracle-head baselines.**
>
> To the best of our knowledge, both MoErging baselines we evaluate use top-1 routing and do not involve temperature scaling or logit calibration. As a result, applying temperature scaling would not meaningfully affect their predictions or the relative comparison.
>
> We share the reviewer’s interest in assessing their performance when equipped with oracle classification heads, as done in their original setting. Below we report the accuracies of the best-performing MoErging baseline (SMILE) against MASS, both using oracle heads for a fair comparison:
>
> | Method | B32 – 8 tasks | B32 – 14 tasks | B32 – 20 tasks | B16 – 8 tasks | B16 – 14 tasks | B16 – 20 tasks |
> | --- | --- | --- | --- | --- | --- | --- |
> | **SMILE (oracle head)** | 87.7 | 85.9 | 85.0 | 92.8 | 92.0 | 91.4 |
> | **MASS (oracle head)** | **89.3** | **86.6** | **86.2** | **93.7** | **92.8** | **91.5** |
>
> As can be seen, MASS maintains a clear advantage even when SMILE is evaluated under its most favorable oracle-head setting. This suggests that the gains of MASS do not only appear in the generalist setup, but are also present in the standard merging setup. We will add these results to the revised manuscript.

---

> ### Author Response · Authors · 2025-11-21
>
> **W5 — Replacing $L_2$ with Mahalanobis distance**
>
> We evaluated a data-free Mahalanobis distance by whitening the task-specific bases derived from the right singular vectors used for projection. All statistics are computed solely from the learned bases, preserving the data-free (including unlabeled) assumption.
>
> **Mahalanobis results**
>
> Given the right singular vectors  $(V_i \in \mathbb{R}^{R \times D})_{ (i = 0,\dots,T)}$ , with $R$ chosen such that $R \times T = D$ , we concatenate and center them: $V_c = V - \mu_V ,\;\mu_V \in \mathbb{R}^{R},$ and compute the covariance: $\Sigma = \mathrm{cov}(V_c) \in \mathbb{R}^{D \times D}.$ The Mahalanobis distance between $x$ and $y$ is then: $d(x,y) = \sqrt{(x-y)^\top \Sigma^{-1} (x-y)}.$
>
> |  | B32 – 8 tasks | B32 – 14 tasks | B32 – 20 tasks |
> | --- | --- | --- | --- |
> | **Mahalanobis** | 95.7 | 91.0 | 88.0 |
> | **L2 (i.e., Ours)** | 96.5 | 93.2 | 90.9 |
>
> Whitening the bases and replacing $L_2$ with Mahalanobis does not improve performance; results are consistently slightly worse, indicating that the Euclidean residual is already sufficiently well-conditioned for our routing mechanism.
>
> **W6 — Cosine similarity vs principal angles for redundancy filtering.**
>
> We thank the reviewer for suggesting this more principled variant. We implemented the proposed approach by computing the principal angles between pairs of task matrices $\Delta_i$. For each candidate $\Delta_i$, we used the largest singular value (equivalently, the minimum cosine between the subspaces) as the similarity measure when deciding whether the update was already “represented’’ by previously accepted ones. As with the cosine-based criterion, we performed a small grid search over the threshold on the 8-task benchmark. We report the normalized accuracies below.
>
> | Method | B32 – 8 tasks | B32 – 14 tasks | B32 – 20 tasks |
> | --- | --- | --- | --- |
> | principal angles | **96.8** | 92.7 | 86.8 |
> | cosine | 96.5 | **93.2** | **90.9** |
>
> The principal-angle criterion slightly outperforms cosine similarity on the 8-task benchmark, but loses accuracy on the larger 14- and 20-task settings. This suggests that while more geometrically principled, the principal-angle filtering is less robust when task diversity increases. We will extend this experiment and include the full results in the revised manuscript.
>
> **W7 — Seeds, variances and wall-clock time.**
>
> Seeds are provided in the supplementary material with the code. We provide the following table with the task-wise mean and standard deviation for the MoErging experiments. In this way is easier to see that MASS has a better average and a smaller standard deviation with respect to the other methods, significantly improving over existing MoErging techniques across all benchmarks:
>
> **ViT-B-32**
> | Method | 8-task | 14-task | 20-task |
> | --- | --- | --- | --- |
> | MASS (Ours) | 96.54 ± 3.94 | 93.23 ± 9.02 | 90.90 ± 9.61 |
> | SMILE-1 | 92.14 ± 5.02 | 84.56 ± 12.63 | 82.32 ± 20.31 |
> | SMILE-2 | 93.58 ± 4.67 | 85.70 ± 13.11 | 83.86 ± 20.56 |
> | WeMoE | 97.74 ± 2.19 | 82.83 ± 28.78 | 76.36 ± 33.45 |
>
> **ViT-B-16**
> | Method | 8-task | 14-task | 20-task |
> | --- | --- | --- | --- |
> | MASS (Ours) | 98.01 ± 1.33 | 96.17 ± 5.63 | 88.74 ± 19.89 |
> | SMILE-1 | 94.92 ± 3.78 | 89.56 ± 10.18 | 86.75 ± 15.29 |
> | SMILE-2 | 96.20 ± 2.83 | 90.70 ± 10.42 | 87.77 ± 15.33 |
> | WeMoE | 96.43 ± 4.72 | 83.26 ± 28.99 | 70.54 ± 32.51 |
>
> **ViT-L-14**
> | Method | 8-task | 14-task | 20-task |
> | --- | --- | --- | --- |
> | MASS (Ours) | 98.68 ± 1.43 | 97.39 ± 4.63 | 96.65 ± 4.25 |
> | SMILE-1 | 96.78 ± 2.90 | 92.78 ± 10.73 | 90.52 ± 13.72 |
> | SMILE-2 | 97.63 ± 2.38 | 93.42 ± 10.93 | 91.15 ± 13.93 |
> | WeMoE | 94.29 ± 7.50 | 72.51 ± 35.64 | 61.41 ± 32.91 |
>
> We report in the following the wall-clock time incurred in a single step.
> |  | TSV | MASS | MASS (no ortho) |
> | --- | --- | --- | --- |
> | per-step wallclock (s) | 0.0031 | 0.013 | 0.0088 |
>
> As can be seen from the results MASS incurs a ~4x increase in latency due to its double forward pass and on-the-fly merging. However, upon further investigation, we found the main bottleneck to be the orthogonalization performed on the adaptive merging step: removing this one almost halves the latency of MASS. Luckily, we have already seen in the answer to W3 that this step actually provides only marginal benefit. Intuitively, interference reduction becomes less crucial when the set of tasks shrinks after the routing step. We therefore thank the reviewer for suggesting the ablation, as this variant will become the suggested one for latency sensitive applications.

---

> ### Author Response · Authors · 2025-11-22
>
> **W7 -- GLUE per-task analysis.**
>
> Thank you for noting this. We now explicitly highlight the per-task comparison in the text: MASS obtains the highest absolute accuracy on 6 out of the 8 GLUE tasks (CoLA, MRPC, QQP, RTE, SST-2, STS-B), while performing slightly below the best baseline on the remaining ones (MNLI, QNLI). We have added a short sentence summarizing these per-task wins/losses next to the table for clarity.
>
> We observe that the two tasks where MASS does not achieve the top score are the NLI-style benchmarks (MNLI and QNLI). This is consistent with these tasks requiring more broad semantic reasoning rather than localized feature shifts, possibly resulting in more diffuse and higher-rank task vectors compared to other GLUE tasks.
>
> **References**
>
> [1] Gargiulo, Antonio Andrea, et al. "Task singular vectors: Reducing task interference in model merging." Proceedings of the Computer Vision and Pattern Recognition Conference. 2025.

---

### Official Review · Reviewer_9j9B · 2025-10-31

**Soundness:** 3
**Presentation:** 3
**Contribution:** 1
**Rating:** 6
**Confidence:** 3

**Summary:**

The paper introduces **MASS**, a novel approach for model merging. It targets the problem of merging multiple fine-tuned models into a single, shared model while retaining performance comparable to that of individually fine-tuned models, without the need for additional training or data. MASS leverages low-rank decomposition and adaptive routing mechanisms to select the most relevant task-specific subspaces for each input, making it an efficient solution for multitask learning scenarios. Experimental results show that MASS achieves near-state-of-the-art performance on various benchmarks while reducing storage requirements compared to traditional ensemble methods.

**Strengths:**

1. **Innovative Concept:** The adaptive routing mechanism for task-specific subspace selection and the use of low-rank decomposition is an interesting approach to reducing storage requirements in model merging.

2. **Practical Application:** The method’s ability to merge models without additional data or retraining is a valuable contribution, especially for scenarios involving pre-trained models from public repositories.

3. **Strong Experimental Results:** MASS performs well across multiple vision and language benchmarks, surpassing existing methods in accuracy and storage efficiency.

4. **Efficiency:**  MASS performs well with a moderate increase in storage compared to a single model, providing a storage-efficient alternative to traditional ensembling methods.

**Weaknesses:**

1. **Generalization to Unseen Tasks:** The approach seems to focus on tasks seen during fine-tuning, but its ability to generalize to unseen tasks or tasks that require more nuanced adaptations is unclear. The task subspaces may not adequately cover unseen task distributions, affecting its robustness in dynamic or changing environments.

2. **Scalability to Larger Task Sets:**  While MASS performs well on 8–20 tasks, its scalability to a larger number of tasks is unclear. As the task set increases, the complexity of the routing mechanism may grow exponentially, making the method less feasible for large-scale applications.

3. **Inference Overhead:** The two-pass inference process introduces additional overhead, which could be problematic in real-time applications where latency is critical.

**Questions:**

1. How does the two-pass inference process impact latency, especially in time-critical applications?
2. How can the method be generalized to handle tasks that were not part of the fine-tuning phase?
3. As the number of tasks increases, how does the method maintain performance and efficiency?

---

> ### Author Response · Authors · 2025-11-21
>
> We thank the reviewer for their comments. We are glad they found the method innovative, practical and efficient.  We address their concerns below.
>
> **W1, Q1 — Generalization to unseen tasks.**
>
> We thank the reviewer for pointing this out. To evaluate how well the method behaves on tasks not included during merging, we conducted an out-of-distribution experiment on the 8-task benchmark. For each task in turn, we removed it from the merge set, merged the remaining seven tasks, and then evaluated on the held-out one. The table below reports the results, comparing MASS with Task Arithmetic:
>
> |  | DTD | MNIST | GTSRB | SVHN | EuroSAT | RESISC45 | Cars | SUN397 |
> | --- | --- | --- | --- | --- | --- | --- | --- | --- |
> | **TA** | 0.3154 (0.4034) | 0.7979 (0.8023) | 0.2671 (0.2700) | 0.4966 (0.5137) | 0.3104 (0.3127) | 0.3692 (0.3874) | 0.4062 (0.5089) | 0.4437 (0.5925) |
> | **MASS** | **0.3691 (0.4721)** | **0.8657 (0.8705)** | **0.3307 (0.3343)** | **0.5970 (0.6175)** | **0.4093 (0.4123)** | **0.4635 (0.4863)** | **0.4946 (0.6197)** | **0.5477 (0.7315)** |
>
> Across all eight held-out tasks, MASS consistently outperforms the standard merging baseline. This suggests that the task subspaces extracted from weight updates contain transferable structure that remains useful even when the target task was never seen during merging.
>
> **W2, Q3 — Scalability to larger task sets.** We thank the reviewer for raising this point. To show that MASS scales beyond 20 tasks, **we collected 13 additional** publicly available datasets to obtain **the largest task set in the model merging literature**. We note that existing model merging approaches either only test on 8 tasks, or also consider the 14- and 20-task benchmarks. The only exception we are aware of is EMR-Merging [1], evaluated on 30 tasks.
>
> We show below that our approach effectively scales up to 33 tasks. We report task arithmetic as a reference: even for 33 tasks, MASS outperforms this baseline by about 20 accuracy points. We also revised the paper by adding a scaling plot showing how the approach behaves when merging $[2, \dots, 33]$ tasks, totalling 560 model evaluations for both MASS and task arithmetic. The plot shows that the performance gap between MASS and task arithmetic remains stable as the number of merged tasks increases.
>
> |  | n=20 | n=25 | n=30 | n=33 |
> | --- | --- | --- | --- | --- |
> | Task Arithmetic | 0.72 | 0.63 | 0.64 | 0.64 |
> | MASS (Ours) | **0.91** | **0.85** | **0.83** | **0.83** |
>
> The method preserves efficiency because the overall compute cost remains dominated by two forward passes, which are independent of the number of merged tasks. The only component that scales with the number of tasks is the routing step. Routing requires projecting the intermediate activation onto each task subspace, which is implemented as a small matrix product per task. This makes the routing complexity **linear in the number of tasks**, hence scaling gracefully.
>
> **W3, Q3 — Inference overhead.**
>
> It is true that the approach requires two forward passes. This is one of the key assumptions of the work: the first pass is necessary to route both the encoder subspace and the correct classification head, and this is precisely what allows the method to approach fine-tuned performance while still producing a single generalist model. Without this initial pass, the merged model would collapse to fixed, non-adaptive merging and we would not be able to identify the appropriate task head.
>
> We also acknowledge that this overhead makes the method unsuitable for some latency-critical scenarios. In batch-based settings, the cost can be amortized by performing the routing once for the entire batch and applying the selected merged model to all samples, which reduces the per-sample overhead substantially. Still, the total latency remains higher than a single forward pass. We report in the following the average wall-clock time incurred by MASS and the fixed merging baseline TSV.
>
> |  | TSV | MASS | MASS (no ortho) |
> | --- | --- | --- | --- |
> | per-step wallclock (s) | 0.0031 | 0.013 | 0.0088 |
>
> As can be seen from the results MASS incurs a ~4x increase in latency due to its double forward pass and on-the-fly merging. However, upon further investigation we found the main bottleneck to be the orthogonalization performed on the adaptive merging step. We therefore also evaluated a non-orthogonalized variant and obtained the following results:
>
> | adaptive merging method | B32 – 8 tasks | B32 – 14 tasks | B32 – 20 tasks |
> | --- | --- | --- | --- |
> | non-ortho | 95.3 | 93.0 | 90.7 |
> | ortho | 96.5 | 93.2 | 90.9 |
>
> | adaptive merging method | B16 – 8 tasks | B16 – 14 tasks | B16 – 20 tasks |
> | --- | --- | --- | --- |
> | non-ortho | 97.3 | 96.1 | 88.5 |
> | ortho | 98.0 | 96.1 | 88.7 |

---

> ### Author Response · Authors · 2025-11-21
>
> | adaptive merging method | L14 – 8 tasks | L14 – 14 tasks | L14 – 20 tasks |
> | --- | --- | --- | --- |
> | non-ortho | 97.8 | 97.3 | 96.6 |
> | ortho | 98.6 | 97.3 | 96.6 |
>
> As we can see, this is in the same ballpark of the orthogonalized variant, especially when scaling the number of tasks, but actually halves the latency incurred by MASS. Intuitively, given that only a small subset of tasks is merged after routing, interference is already limited, so full orthogonalization becomes less critical. For this reason, we argue that for latency-critical applications the latter might be preferred.
>
> We believe other optimizations to be possible, and view this as an opportunity for future work: the present framework establishes a principled way to route through task subspaces and heads, and subsequent research may build on this direction to develop lighter, more efficient routing and merging mechanisms.
>
> **References**
>
> [1] Huang, Chenyu, et al. "Emr-merging: Tuning-free high-performance model merging." *Advances in Neural Information Processing Systems* 37 (2024): 122741-122769.

---

### Author Response · Authors · 2025-12-01
**Author Response Summary for the Area Chair**

Dear AC and Reviewers,

We sincerely thank all reviewers for their effort. We are glad the method was judged innovative (`9j9B`), practical and impactful (`9j9B`), simple yet effective (`eoaD`), and grounded in a realistic problem setting (`JZ1o`).

To support the Area Chair, we briefly summarize the main concerns raised and how we addressed them in the rebuttal.

- Reviewer `9j9B` raised concerns about OOD generalization, scaling to a large number of tasks, and latency overhead due to the double forward pass. For **OOD generalization**, we added a new experiment on the 8-task benchmark: for each task, we removed it from the merge set, merged the remaining seven, and evaluated on the held-out one. MASS outperforms Task Arithmetic on all eight OOD tasks. To show scaling, we **added 13 additional tasks** (up to 33 tasks total), yielding the largest benchmark in the merging literature and showing that MASS maintains a large accuracy margin as the number of tasks grows. Finally, we reported **wall-clock times** for TSV and MASS, identifying orthogonalization in adaptive merging as the main bottleneck; removing it roughly halves latency while preserving accuracy, giving a **low-latency variant** that still reaches state-of-the-art performance.  While the discussion was halted before the reviewer could follow up, we believe the new experiments and clarifications comprehensively address their concerns.
- Reviewer `8SY3` questioned the “data-free” framing, the design choices in routing and redundancy filtering, and the lack of variance and timing. We clarified that **routing layers are chosen once** and are stable across tasks, and that **thresholds are tuned only on the 8-task benchmark** and reused for 14- and 20-task setups. We implemented and reported several requested ablations: a **weighted TSV variant** (no gain over α=1.0), **orthogonalization vs non-orthogonalization** in adaptive merging (small but consistent benefit with orthogonalization), Mahalanobis vs L2 residual (L2 performs better), **cosine vs principal-angle** redundancy filtering (principal angles help at 8 tasks but hurt at 14/20), and an **oracle-head comparison** where SMILE uses its best setting and MASS still performs better. We also added per-task mean/variance tables and wall-clock measurements. Although the discussion ended before the reviewer could revisit their points, the ablations and additional analyses we provided directly resolve the issues they raised.
- Reviewer `eoaD` was mainly concerned with the clarity of Section 3.2.1 and the reasonableness of assuming unknown task identity at inference. We revised the projector definition and residual notation to make the section precise and unambiguous. We clarified the setting: while the fine-tuning task of each checkpoint is known, the **task of each input sample at inference** is not, which is exactly the realistic regime we target. Since the discussion stopped before the reviewer could respond further, the clarifications we introduced should fully clear up the misunderstandings underlying their concerns.
- Reviewer `JZ1o` questioned the incrementality of the contribution and the lack of experiments on decoder-only LLMs and generative tasks. We clarified that in MoErging the core novelty typically lies in routing/selection rather than in re-defining the aggregation, and that our main contribution is a **training- and data-free router and head selector** operating solely in weight space. On scope, we explained that our work targets the explicit task boundary regime (vision + encoder-only language models) where preserving each expert’s accuracy is the goal, while decoder-only LLM merging follows a different, compositional regime with distinct methods and objectives. We also reported **13 prior works** published in top AI conferences following this exact setting, all of which evaluate on benchmarks that are subsets of the benchmarks we consider. We remark that considering our evaluation scope insufficient because of lacking decoder-only LLMs would effectively call into question the adequacy of this entire research line as practiced in the literature. To further address the reviewer’s concern, we additionally **merged three Mistral-7B finetunings** and found that MASS outperforms both the experts and standard TA on three out of four LLM benchmarks, indicating that the method extends to LLMs.

To summarize, reviewers `9j9B` and `8SY3` requested concrete experiments, all of which we provided and which we believe fully address their concerns. Reviewer `eoaD` asked only for clarifications, which we supplied. Reviewer `JZ1o` raised opinion-based concerns regarding incrementality and the absence of large-scale decoder-only LLM experiments. We first discussed the sufficiency of the evaluation setting, and we then provided results on the latter models, effectively addressing the only actionable point raised.

We thank all reviewers and the Area Chair for their time and consideration.

The Authors

---

### Meta-Review · Area_Chair_WGhJ · 2026-01-07

**Summary:**

The reviewers raised concerns around robustness, scalability, routing design choices, clarity of the formulation, and the scope of evaluation. The rebuttal substantively addressed most technical questions with additional experiments, ablations, and clarifications, including OOD evaluation, scaling to a large number of tasks, latency analysis, and detailed design justifications. These responses resolve the main technical doubts raised by three reviewers.

The remaining concern is about perceived incrementality and limited evaluation on large decoder-only LLMs, raised by one reviewer, who explicitly maintained their original score. While this limits the perceived breadth of impact, the scope and problem setting are consistent with a well-established line of work in model merging, and the added evidence supports the paper’s claims within that regime.

Overall, the paper presents a solid, well-executed contribution with strong empirical support for its intended setting, leading me to recommend weak accept.

**Reviewer Concerns:**

Reviewer Concerns

Reviewer 9j9B:
The main technical concerns (OOD generalization, scalability, and inference overhead) were addressed with new experiments and measurements. While latency remains a practical trade-off, the rebuttal provides enough evidence that the method scales and behaves robustly. No major unresolved issues.

Reviewer 8SY3:
Most of the reviewer’s concerns were directly addressed through additional ablations and analysis (routing design choices, orthogonalization, weighting, distance metrics, oracle-head comparisons, variance, and timing). The only remaining point is a minor tension around the “data-free” framing, but this does not undermine the core technical claims.

Reviewer eoaD:
Concerns were mainly about clarity and the reasonableness of the problem setting. The rebuttal clarifies the notation and explains the unknown-task-at-inference assumption clearly. These issues are resolved.

Reviewer JZ1o:
The reviewer’s concerns about incrementality and the lack of large-scale decoder-only LLM experiments remain. Although the rebuttal clarifies scope and adds a small LLM experiment, the reviewer explicitly maintained their original position, so these concerns should be considered outstanding.

**Reviewer Scores:**

Reviewer Scores (expected after full discussion)

Reviewer 9j9B: 6 → 6
The rebuttal adds solid new experiments (OOD hold-out, scaling to 33 tasks, and latency numbers). That said, the reviewer didn’t explicitly indicate a score change, and latency is still a real trade-off. So I’d keep them at 6.

Reviewer 8SY3: 4 → 6
This reviewer asked for specific ablations/analyses (variance, timing, weighted variant, orthogonalization, Mahalanobis vs L2, redundancy filtering, oracle heads). The rebuttal directly responds with comprehensive results and clarifications, and it’s reasonable to expect the main technical doubts are largely cleared. I’d move to 6.

Reviewer eoaD: 4 → 6
Their concerns were mostly clarity + justification of the “unknown task at inference” setting. The rebuttal makes the definitions precise and the setting argument is convincing. With those resolved, a bump to 6 seems reasonable.

Reviewer JZ1o: 4 → 4
The reviewer explicitly stated they will maintain their original score after reading the rebuttal. Even with the added small LLM experiment and scope clarification, I don’t see grounds to override that. Keep at 4.

Summary: 6 / 6 / 6 / 4 → Average = 5.5

---

### Decision · Program_Chairs · 2026-01-26

Accept (Poster)